# Drought recovery in plants triggers a cell-state-specific immune activation

Natanella Illouz-Eliaz[1,2,3], Jingting Yu [4], Joseph Swift[1,2], Kathryn Lande[4,5], Bruce Jow[1,2,3], Lia Partida-Garcia[1,3], Za Khai Tuang [6], Travis A. Lee [1,2,3], Adi Yaaran[7], Rosa Gomez-Castanon [1,2], William Owens[2], Chynna R. Bowman[8], Emma Osgood[2], Joseph R. Nery [1,2], Tatsuya Nobori [1,2,9], Yotam Zait[7], Saul Burdman [6] & Joseph R. Ecker [1,2,3] ✉

All organisms experience stress as an inevitable part of life, from single-celled microorganisms to complex multicellular beings. The ability to recover from stress is a fundamental trait that determines the overall resilience of an organism, yet stress recovery is understudied. To investigate how plants recover from drought, we examine a fine-scale time series of RNA sequencing starting 15 min after rehydration following moderate drought. We reveal that drought recovery is a rapid process involving the activation of thousands of recovery-specific genes. To capture these rapid recovery responses in different *Arabidopsis thaliana* (*A. thaliana*) leaf cell types, we perform a single-nucleus transcriptome analysis at the onset of drought recovery, identifying a cell type-specific transcriptional state developing independently across cell types. To further validate the cell-type specific transcriptional changes observed during drought recovery, we employ spatial transcriptomics using multiplexed error-robust fluorescence in situ hybridization (MERFISH), revealing anatomical localization of recovery-induced gene expression programs across Arabidopsis leaf tissues. Furthermore, we reveal a recovery-induced activation of the immune system that occurs autonomously, and which enhances pathogen resistance in vivo in *A. thaliana*, wild tomato (*Solanum pennellii)* and domesticated tomato (*Solanum lycopersicum* cv. M82). Since rehydration promotes microbial proliferation and thereby increases the risk of infection, the activation of drought recovery-induced immunity may be crucial for plant survival in natural environments. These findings indicate that drought recovery coincides with a preventive defense response, unraveling the complex regulatory mechanisms that facilitate stress recovery in different plant cell types.

In most plants, extended periods of water deficit result in reduced growth, premature flowering, flower abortion, fruit abscission, and, ultimately, decreased yield[1,2]. Plant responses to drought have therefore been studied extensively as part of efforts to develop strategies or genetic manipulations that could mitigate the economic and agricultural consequences of future droughts.

Previous studies have identified and functionally characterized numerous genes that respond to water deficit, including various transcription factors (TFs) that regulate plant drought responses[2–4]. In general, drought stress in plants induces dramatic changes in the transcriptional landscape[5,6]. For example, the rapid up-regulation of genes involved in osmolyte accumulation enables water retention by

adjusting cellular osmotic potential[7]. However, efforts to enhance drought tolerance through genetic manipulation have frequently resulted in undesired growth inhibition even under non-drought conditions, thereby constraining the widespread engineering and adoption of drought-resilient crops[4,8,9].

Given these limitations, we considered an alternative approach to investigating stress resilience in plants by focusing on post-drought recovery rather than drought resistance. Indeed, understanding a plant's ability to recover from drought is central to a comprehensive understanding of drought resilience, as the potential for recovery defines whether the system can return to a stable state of function[10]. For example, drought recovery has been established as an indicator of drought tolerance in annual crops such as maize[11,12], wheat[13,14], and rice[15,16]. This extends to other abiotic stresses. For instance, the rates of submergence recovery in *Arabidopsis thaliana* (Arabidopsis) accessions were found to correlate with their submergence tolerance[17]. This finding suggests that the ability to recover after flooding is critical for plant survival and reproductive success. One of the well-characterized plant responses to drought alleviation is the downregulation of the drought-responsive genes. It was previously shown that most drought-regulated genes recover to normal expression levels within three hours of rehydration[18]. However, Oono et al.[19] identified 82 "recovery-specific" genes whose expression was drought-invariable but altered by subsequent rehydration.

Despite these known transcriptional responses during drought recovery, it remains unclear whether these responses constitute a conserved drought recovery mechanism activated throughout the entirety of plant cell types. Here, we explore the transcriptomic landscape in Arabidopsis throughout the drought recovery process using a high-resolution time series of RNA sequencing data. We identified over 3000 recovery-specific genes that were differentially expressed between 15 min and 6 h after rehydration. Using single-nucleus RNA-seq, we analyzed cell type-specific transcriptional signatures immediately after inducing drought recovery. We identified a recovery unique transcriptional signature common between sub-population of epidermal, hydathode, sieve element, and mesophyll cells that is found only once drought recovery is initiated. Within the gene modules enriched in sub-populations, we found many immune-associated genes. Based on these observations, we propose that some cells enter a recovery cell state upon rehydration. The results indicated that one of the first steps of the recovery process is the activation of a rapid preventive immune response. We therefore tested whether short-term recovery from moderate drought activated functional immunity to block pathogen proliferation in plant leaves. In agreement with this hypothesis, we found that drought recovery-induced activation of immune system genes leads to reduced disease severity and bacterial load in infected leaves of both Arabidopsis and two tomato species.

## Drought recovery-specific genes

To study the transcriptional response to drought recovery, we performed a fine temporal resolution RNA-seq using vermiculite-grown Arabidopsis rosettes. We collected Arabidopsis rosettes during moderate drought (defined by the relative soil water content reduced to 30%) and at seven subsequent recovery time points within the six hours following rehydration (Fig. 1a, b). At each time point, well-watered rosettes were also collected to control for diurnal transcriptional changes (Fig. 1 b). We then performed a differential gene expression analysis across treatments and time points. Based on this analysis, we defined two groups of genes: drought-responsive genes and recovery-specific genes. Drought-responsive genes were defined as genes that were differentially expressed (DE) in the drought-treated plants (time 0) compared to well-watered plants collected at the same time (Supplementary Data 1). Conversely, recovery-specific genes were not DE during the drought time point that was collected, but were DE in any of the post-drought recovery time points (Supplementary Data 2).

Based on these definitions, we identified 1248 drought-responsive genes, of which 662 (53%) were also DE during the recovery time points that were sampled. After only six hours of rehydration, 97% of these genes returned to normal expression levels (Fig. 1c, d). Additionally, we identified over 3000 recovery-specific genes across all recovery time points (Fig. 1c, e), suggesting that recovery involves regulatory programs beyond mere reversal of drought-induced changes. We used K-means clustering to characterize gene expression patterns during drought and recovery. The drought-responsive genes exhibited five distinct expression patterns: (1) drought-induced but downregulated during recovery, (2) drought-induced but not DE during recovery, (3) drought- and recovery-induced; (4) drought-downregulated but recovery-induced; and (5) drought-downregulated but not DE during recovery (Fig. 1d). Recovery-specific genes exhibited three types of expression patterns based on whether they were induced in (1) early recovery or (2) late-recovery, or (3) downregulated by rehydration (Fig. 1e). These results demonstrate that drought recovery involves the activation of thousands of recovery specific genes.

## Rapid recovery of physiological responses

To test whether the transcriptional temporal dynamics revealed by RNA-seq are observable *in planta* we conducted physiological experiments to assess the water status within leaves. We measured the relative water content (RWC) of leaf 6 from each rosette under four conditions: well-watered, moderate drought, 15 min of rehydration, and 60 min of rehydration. Our results indicate reduced leaf RWC during moderate drought, with no significant change in leaf water content after 15 min of rehydration. After 60 min a significant increase in the RWC compared to drought-treated plants was observed (Fig. 2a).

To further investigate, we examined stomatal conductance to water vapor (Fig. 2 b) using a leaf porometer. Stomatal conductance significantly recovered toward well-watered levels after just 15 min of rehydration, with complete restoration of the stomatal conductance observed after 60 min. Notably, this rapid recovery of stomatal conductance occurred independently of the slower changes in leaf RWC. These findings confirmed that under moderate drought stress, leaf water content and stomatal conductance are reduced, and revealed that the stomatal conductance responded to rehydration more quickly, achieving complete recovery within 1 hour.

## Global and cell type-specific recovery induced transcriptional reprogramming

To study the leaf cell type-specific transcriptional responses upon drought recovery, we performed single-nucleus RNA-seq (snRNA-seq). For this experiment, we processed two replicates undergoing long-term moderate drought and an additional two replicates after 15 min of rehydration, as well as equivalent samples from well-watered controls. In this case, the well-watered controls were also provided with 15 min of additional irrigation to ensure that any changes in gene expression were specific to drought recovery and not due to root mechanical stimulus (Fig. 3a). After data quality control and filtering, our integrated dataset included over 144,000 single-nuclei transcriptomes. The median number of unique molecular identifiers (UMI) per nucleus was between 2534 to 3995 across the eight samples, and the median number of genes detected per nucleus was between 1213 to 1698 (average 1530; see Supplementary Data 3).

Unsupervised clustering of the snRNA-seq profiles resulted in 27 clusters representing distinct molecular cell identities (Supplementary Figs. 1, 2). All independent samples and treatment conditions integrated into the same clusters (Supplementary Fig. 1a–d). To annotate the clusters, we curated a literature-based list of genes with cell type- or tissue-specific expression (Supplementary Data 4), and examined their expression in the top 50 cluster marker genes in our dataset (Supplementary Data 5).

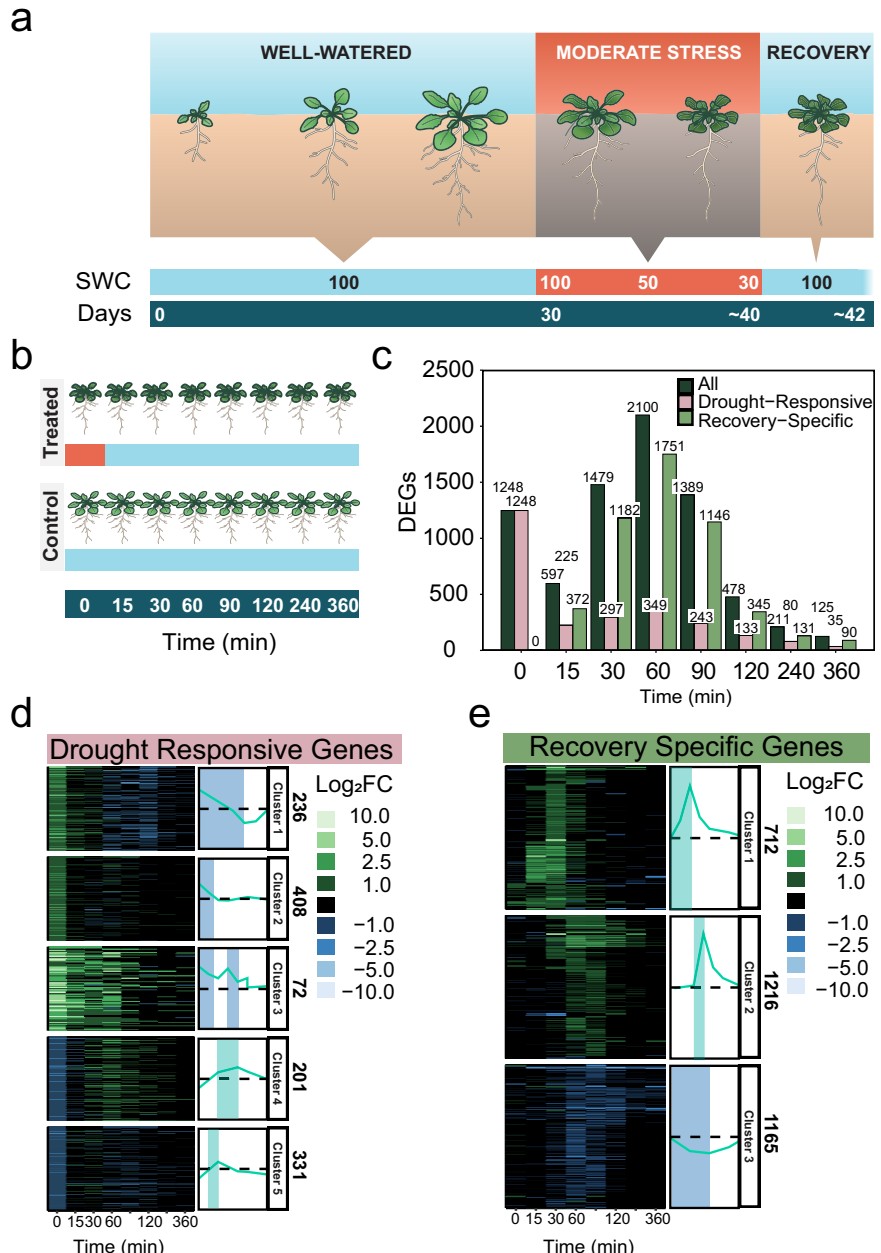

**Fig. 1 | A fine-scale RNA-seq time course of drought recovery reveals recovery-specific genes. a** Illustration of the recovery time-course experimental design. Plants were grown in vermiculite under a short-day photoperiod. After 30 days, irrigation was stopped for drought-treated plants until they reached 30% relative soil water content (SWC). We then rehydrated the drought-treated plants by saturating the vermiculite to initiate drought recovery. We collected 3 biological samples from each time point of recovery, and well-watered control at each time point ($n = 3$). **b** Illustration of the samples collected for RNA-seq. We collected samples during moderate drought ($t = 0$ at 30% SWC) and at seven additional time-points during the recovery process, from 15 min to six hours after rehydration. All samples from drought-treated plants were collected alongside equivalent samples from well-watered controls, with three replicates per treatment (drought or control) per time point. **c** Number of differentially expressed genes (DEGs) at different time points during recovery. For each time point, we show the total number of DEGs as well as the number of drought-responsive and recovery-specific genes. DEGs were identified by comparing drought-treated samples to well-watered controls at each time point. K-means clustering and expression patterns of (**d**) drought-responsive and (**e**) recovery-specific genes.

Epidermal clusters were identified using markers such as *FDH* (*AT2G26250*), a wax biosynthesis gene implicated in epidermal development[20], *LTPG1* (*AT1G27950*), a lipid transfer protein expressed in the epidermis[21], and *PDF1* (*AT2G42840*), a defensin-like protein localized to protodermal cells[22]. Trichome identity was supported by *GL1* (*AT3G27920*) and *GL2* (*AT1G79840*), classical regulators of trichome initiation and development[23–25]. Guard cell clusters were annotated using *GASA9* (*AT1G74670*), a gibberellin-regulated gene expressed in guard cells[26], and *AT1G04800*, recently validated by Tenorio Berrío et al.[27] as a marker of mature guard cells. Myrosin cells expressed *TGG1* (*AT5G26000*) and *TGG2* (*AT5G25980*), enzymes involved in glucosinolate hydrolysis[28]. Mesophyll cells were marked by increased expression of *LHCB2.1* (*AT2G05100*)[29], *RBCS2B* (*AT5G38420*)[29] and *FBA5* (*AT4G26530*)[27]. Phloem companion cell identity was supported by *SUC2* (*AT1G22710*), a sucrose transporter[30], and *NAKR1* (*AT5G48160*), which regulates phloem loading and long-distance signaling[31]. *CDC20-1* (*AT4G33270*) and *CDC20-2* (*AT3G44300*) were more highly expressed in clusters we annotated as cells undergoing division, consistent with their roles in cell cycle progression[32,33] (Supplementary Fig. 3-5). Cluster annotations were

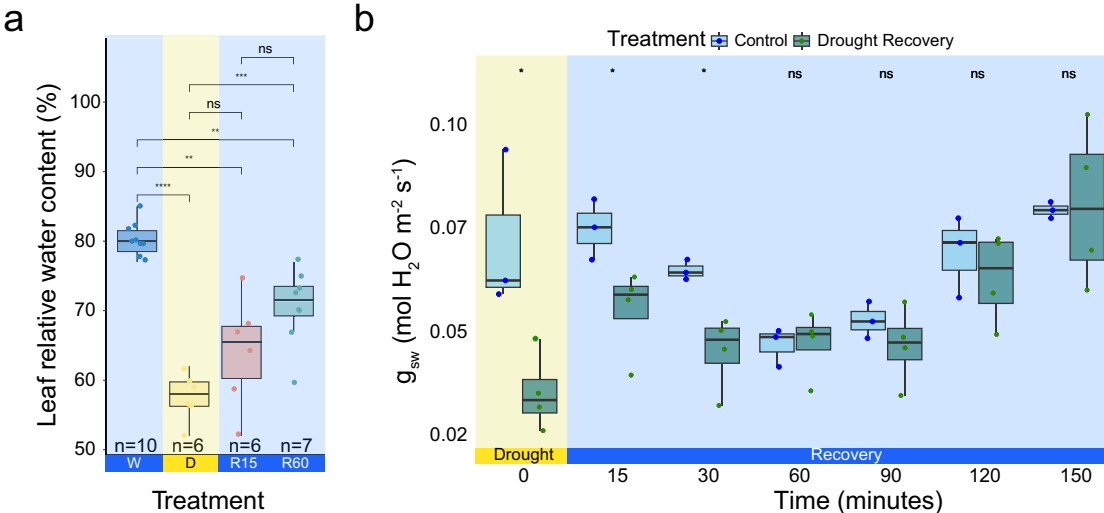

**Fig. 2 | The leaf relative water content and stomatal conductance are partially recovered after 15 mins of rehydration. a** Leaf relative water content of vermiculite-grown plants, comparing well-watered (WW), to moderate drought 30% relative soil water content (D), and drought recovery after 15 mins of rehydration (R15), and 1 h after rehydration (R60). n indicated in the plot for each condition is the number of individual leaves measured. **b** Leaf porometer measurements of the stomatal conductance of plants recovering from moderate drought. Time 0 = 30% SWC, drought, and subsequent time-points are measurements taken after rehydration. Significant differences in panel a were identified using student's t-test (two-sided) and in panel b using Two-Way ANOVA and Tukey-Kramer Post-Hoc test. For well-watered samples $n = 3$, for drought recovered samples $n = 4$. Boxplots middle line shows the median, the lower and upper hinges are the 25th and 75th percentiles, respectively. The whiskers extend from the hinges to the most distant value within 1.5 * IQR of the hinge, where IQR is the inter-quartile range, or distance between the first and third quartiles.

further examined and validated by projecting our cells onto other Arabidopsis leaf single-cell datasets from Procko et al.[34] and Lopez-Anido et al.[35]. Notably, the identification of sieve element cells was only possible after performing the projection-based analysis (Supplementary Fig. 6).

To further confirm our cluster annotations and assess spatial coherence, we performed multiplexed error-robust fluorescence in situ hybridization (MERFISH) using the MERSCOPE platform. We generated spatial maps of gene expression for cell type–specific markers and uncharacterized cluster markers across Arabidopsis leaf cross-sections under all four treatment conditions we sampled in our snRNA-seq (Fig. 3b–d). Spatial localization of cluster markers identified in our snRNA-seq data, spatially aligned with known cell type markers (e.g. *SUC2*[30] for phloem, *ATML1*[36] for young epidermis), and were localized to the cell type predicated by our cluster annotations (Supplementary Figs. 7–9). This high spatial congruence supported the robustness of our cell-type annotations and allowed the identification of novel cell type markers in the Arabidopsis leaves (Supplementary Data 6).

We investigated the top marker genes in each cluster and used Gene Ontology (GO) enrichment analysis to see enriched biological processes (Known cell types; Supplementary Fig. 10). Based on the genes expressed in each cluster and their expression levels, we were able to confidently assign cell types to 22 of the 27 clusters.

To ensure high-confidence annotation, we applied conservative criteria and only assigned cell type identities to clusters that showed minimal to no overlap in marker gene expression across distinct cell types and were enriched for multiple known markers of a single identity. As a result, five clusters remained unannotated, which we designated unknown 1–5. To further characterize the five unannotated clusters, we performed GO enrichment analysis on the top differentially expressed genes (DEGs) from each population. This analysis revealed distinct functional profiles that support the existence of unique, previously unrecognized cell states (Supplementary Fig. 11). Unknown 1 was enriched for immune- and stress-related GO terms, including response to bacterium (GO:0009617), response to oxidative stress (GO:0006979), and immune system process (GO:0002376), suggesting that this population represents an immune active cell state involved in both biotic and abiotic defense responses. Unknown 2 shared many of these immune-related terms but also showed enrichment for defense response to insect (GO:0002213) and abscission (GO:0010060), consistent with a defense state likely involved in wounding, abscission, or tissue remodeling during stress. Unknown 3 displayed a broad, less specialized enrichment pattern with terms such as response to bacterium (GO:0009617) and response to organic cyclic compound (GO:1901360), indicative of a stress responsive cell population with general adaptability to environmental challenges. Unknown 4 exhibited specific enrichment for response to oxidative stress (GO:0006979) and response to toxic substance (GO:0009636), defining it as a metabolic stress state that likely plays a role in detoxification and cellular protection against reactive or harmful compounds. Finally, cluster unknown 5 was associated with terms related to carbohydrate and solute transport, such as monosaccharide metabolic process (GO:0005996), carbohydrate transport (GO:0008643), and fluid transport (GO:0006817), pointing to a sugar metabolic state involved in nutrient allocation and energy metabolism. Spatial mapping of marker genes from the "unknown" clusters revealed some discrete anatomical localization patterns, such as vasculature-localization for cluster "unknown 5", but also overlapping expression across several cell types, suggesting these cells cluster based on a functional cellular state (Supplementary Fig. 12).

The GO enrichment analysis coupled with spatial transcriptomics provides a useful framework for interpreting the potential roles of these unannotated clusters in the well-watered, dehydration, and recovery response (Supplementary Figs. 11, 12). Overall, the snRNA-seq data revealed 10-fold more DEGs than detected by bulk RNA-seq (Supplementary Fig. 13).

Once the clusters had been assigned to cell types, we investigated transcriptional changes during the onset of drought recovery across the Arabidopsis leaf cell types. A transcription factor (TF) expression analysis across the different treatments revealed that TFs are rapidly induced by rehydration across a broad range of cell types (Supplementary Fig. 14, 15). These TFs included many genes from the *ETHYLENE RESPONSE FACTOR* (*ERF*) and *WRKY DNA-BINDING PROTEIN* (*WRKY*) families (Full list in: Supplementary Data 7, statistical analysis

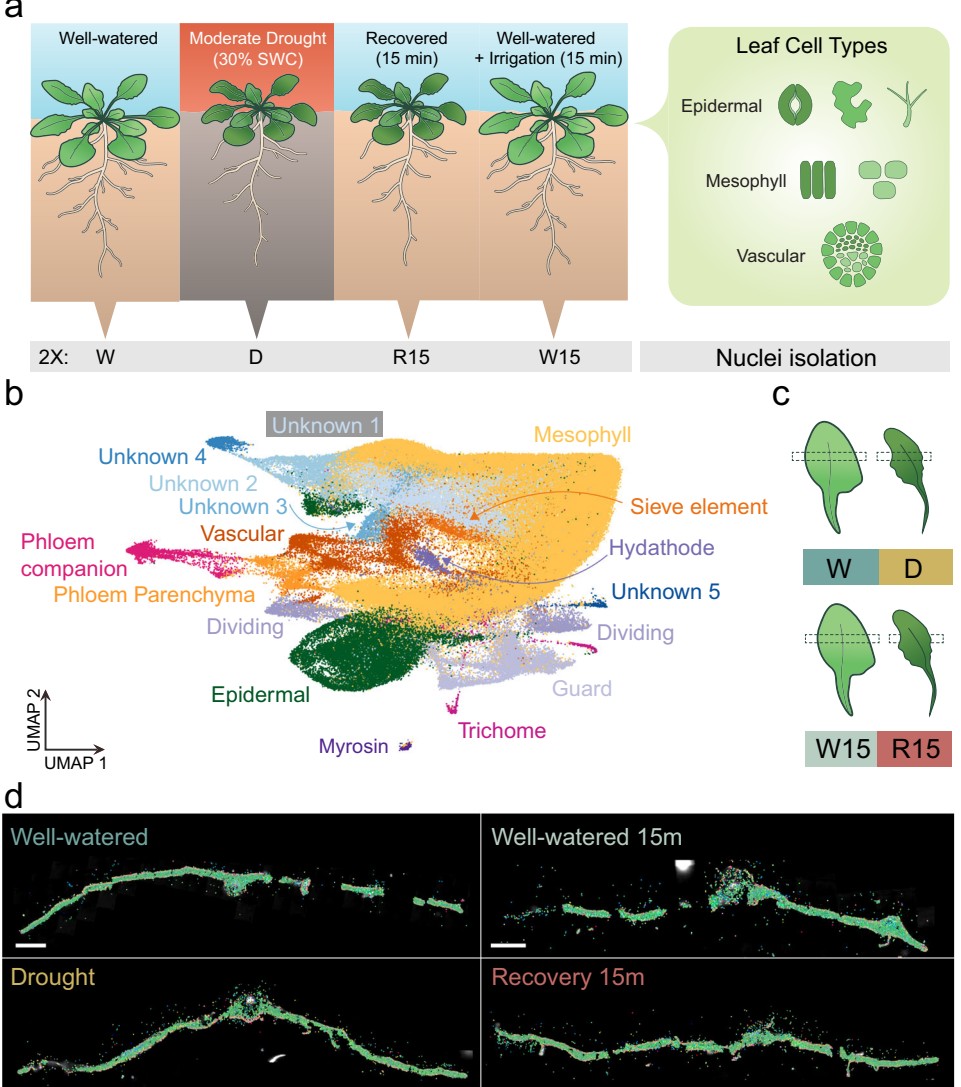

**Fig. 3 | Cell type–resolved transcriptional responses to post-drought rehydration in Arabidopsis leaves.** a Schematic of the experimental design illustrating Arabidopsis plants subjected to four water status conditions: well-watered (W), moderate drought at 30% soil water content (SWC; D), recovery for 15 min following rewatering (R15), and a 15-minute irrigation treatment under well-watered conditions (W15). Leaf nuclei were isolated from Arabidopsis rosette leaves at each condition and analyzed for condition- and cell type–specific responses. b UMAP projection of single-nucleus transcriptomes from all conditions reveals major leaf cell types including mesophyll, epidermal, vascular, phloem (companion and parenchyma), sieve element, hydathode, trichome, myrosin, dividing, and guard cells. c Leaf sampling strategy showing dissection regions used for spatial transcriptomics using MERFISH in all four conditions. d Spatial expression patterns of transcripts of a 1000-gene panel, designed based on snRNA-seq data across the four conditions using the MERFISH platform. Scale bar = 1 mm.

in: Supplementary Data 7, bottom; Supplementary Fig. 15). We also observed a group of TFs that were preferentially upregulated in dividing cells during early recovery (Full list and statistical analysis in: Supplementary Data 8, Supplementary Fig. 14b). Among the most significantly upregulated TFs was *MYB124*, also known as *FOUR LIPS* (*FLP*), which plays a role in stomatal development[37] and has been shown to enhance drought tolerance in apple[38]. Overall, our findings indicate that drought recovery triggers widespread transcriptional activation of TFs, including some that show enhanced expression in specific cellular contexts such as dividing cells.

## Recovery cell states in distinct cell types

To study heterogeneity within cell types during drought recovery, we further subclustered cells within each cell type based on their gene expression profiles (Supplementary Figs. S16–S19). Out of the 16 cell identities identified in our leaf drought and recovery dataset (Fig. 3

b), we observed a recurring pattern across four different cell types. In epidermal, mesophyll, hydathode, and sieve-element cells, we identified cell subpopulations that were highly enriched (i.e., over 50% of cells in the subcluster) with cells from the drought recovered plants (Fig. 4a, Supplementary Fig. 19, 20). We investigated whether these subpopulations within the different cell types expressed similar gene networks by performing gene co-expression network analysis[39] for the seven cell clusters with a recovery enriched subpopulation (Fig. 4). For each subcluster, this analysis identified enriched gene modules (Fig. 4d, e). We next searched for common hub genes (Genes with high module membership (kME) and high intramodular connectivity (kIN)) among the modules that were enriched in these subpopulations across cell types. These hub genes may have key roles in the formation of this cell state or its function (Supplementary Figs. 21, 22). Overall, we found 212 hub genes, of which 50 were shared by at least two modules (Supplementary Data 9).

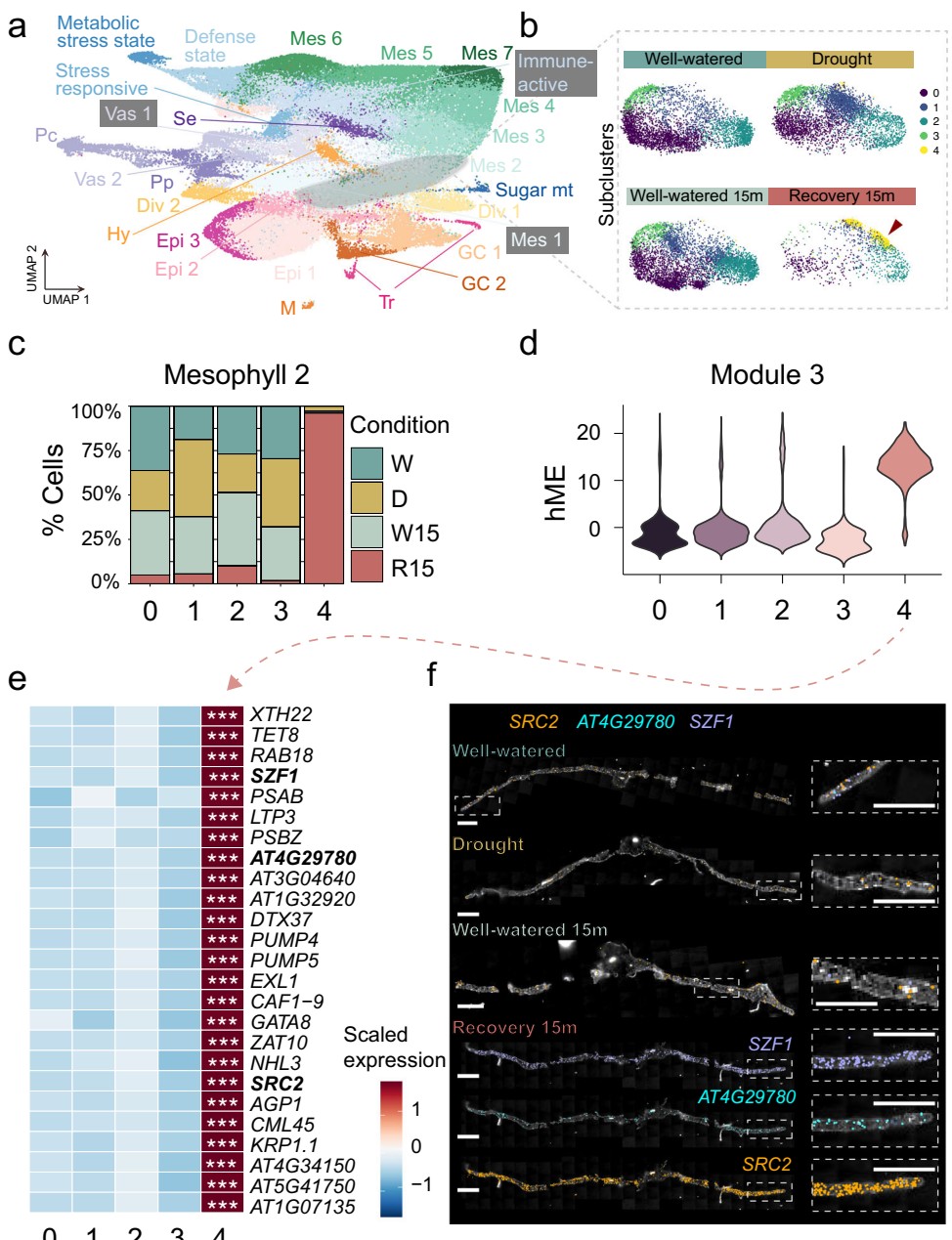

**Fig. 4 | Unique mesophyll subpopulations display a distinct gene expression program during drought recovery. a** UMAP visualization of single-nucleus RNA-seq data from Arabidopsis leaf tissue, showing major cell types including refined cell type clusters. Mesophyll 2 (Mes 2) is selected for subclustering representation. **b** UMAPs of Mes 2 cells partitioned into five subclusters (0–4), colored by subcluster identity by water condition: well-watered (W), drought (D), 15 min post-rewatering (W15), and 15 min into recovery (R15). Subcluster 4 is highly enriched in recovery (R15) (red arrowhead), suggesting a recovery-specific cell state. Sugar mt = Sugar metabolic state. **c** Stacked bar plot showing the distribution of water conditions within each Mes 2 subcluster. Subcluster 4 contains the highest proportion of recovery (R15) nuclei. **d** Violin plots of module 3 abundance (hME score) across Mes 2 subclusters. Subcluster 4 shows elevated expression of this module. **e** Heatmap of module 3 top 25 hub genes, showing specific upregulation in subcluster 4. Asterisks denote genes significantly enriched (FDR < 0.001). **f** Spatial transcriptomic localization of *SRC2*, *AT4G29780*, and *SZF1* across the four water conditions. Signal is enhanced in mesophyll cells, particularly during recovery. Insets highlight localized gene expression. Scale bars = 1 mm.

Interestingly, most subclusters of the immune-active cell state (originally "unknown 1") were enriched in R15 samples. We then examined the overlap between the cluster marker genes of the immune-active cells in this cluster, with the shared hub genes within the different recovery enriched subpopulations in different cell types and found that 70% of them (35/50) overlapped with the above-mentioned hub genes.

To examine the spatial distribution of hub genes across the leaf tissues, spatial transcriptomics was performed on Arabidopsis leaves

harvested under well-watered (0 and 15 min), drought, and recovery (15 min post-rehydration) conditions using a targeted probe panel comprising hub genes identified from our snRNA-seq analysis, along with canonical cell type markers. Notably, several hub genes identified in a mesophyll RcS subpopulation—including *SOYBEAN GENE REGU-LATED BY COLD-2* (*SRC2*), *AT4G29780*, and *SALT-INDUCIBLE ZINC FIN-GER 1* (*SZF1*)—were specifically enriched during early recovery (Fig. 4d–f). The spatial data provide independent validation of the activation of a rehydration induced cell state and offer new insight into

its tissue localization and temporal dynamics during the drought recovery process.

We hypothesized that these cell subpopulations form distinct subclusters due to a transcriptional reprogramming and thus a transition into a recovery cell state (RcS). We examined the expression of the RcS hub genes originally identified from snRNA-seq in our drought recovery bulk RNA-seq time-course (Supplementary Fig. 23). We found that ~34% of RcS activated genes were detectable in bulk RNA-seq after 15 mins of rehydration. Taken together, these finding suggest that drought recovery triggers the formation of a specific cell state imposed within several leaf cell types.

Motif analysis revealed a shared RcS-enriched binding site predicted to be targeted by *CALMODULIN BINDING TRANSCRIPTION ACTIVATOR 1* (*CAMTA1*) and *CAMTA5*, which was found in nearly all RcS subclusters but only rarely in others, suggesting a potential regulatory role (Supplementary 24 a, Supplementary Data 10). In Arabidopsis, the CAMTA family of TFs regulate plant defense and stress responses by modulating the expression of genes involved in pathogen defense, abiotic stress tolerance, and plant development. These TFs interact with calcium-bound calmodulin to mediate downstream signaling pathways[40].

Although *CAMTA* genes were not upregulated at the time of rehydration in our recovery dataset (Supplementary Fig. 25), an early drought time-course snRNA-seq revealed early drought-induced expression of *CAMTA* genes (Supplementary Figs. 26–28), suggesting that *CAMTAs* respond early to drought and may prime the RcS transition. Gene expression analysis of *camta1* mutants during recovery showed no major changes compared to wild-type (Supplementary Fig. 29), however, expression of *CAMTA2–6* was elevated, indicating potential compensatory regulation within the *CAMTA* family (Supplementary Fig. 29f).

We next examined if *CAMTA1's* DAP-seq peaks overlap with promoters of key RcS hub genes. Indeed, we found CAMTA1 motifs in many, such as TETRASPANIN8 (*TET8*) and *SZF1* (Supplementary Fig. 30b). We showed above that *CAMTA* transcripts accumulate in early stages of dehydration (Supplementary Fig. 28), thus CAMTA may inhibit the expression of RcS hub genes during drought and not induce them during recovery. It is possible that upon rehydration, this inhibition is reversed to allow rapid upregulation of recovery-induced genes by another recovery-specific TF. Our data shows that while in wild-type, several RcS hub genes—including *SZF1 (AT3G5S980), PUMP4 (AT4G24570), AT1G32920* and *AT3G04640*—are downregulated during drought relative to well-watered conditions, in the *camta1* mutant, this downregulation is attenuated or absent, implying that *CAMTA1* is required for drought-induced repression of these genes (Supplementary Fig. 30).

Overall, the suite of common hub genes among the six RcS subclusters from different cell types suggests that the functional onset of drought recovery in RcS subpopulations is comprised of cell wall modifications and the regulation of nutrient uptake, as well as cytoplasmic detoxification processes and DNA repair. The most prevalent hub genes shared among RcS subclusters are genes that play a critical role in cell growth and plant development; these genes include *XYLOGLUCAN ENDOTRANSGLUCOSYLASE/HYDROLASE 22* (*XTH22*)[41], *SLAC1 HOMOLOGUE 3* (*SLAH3*)[42], and *EXORDIUM-LIKE1* (*EXL1*)[43]. Since toxins accumulate in plant cells during drought stress[44], detoxification may also be an important part of recovery. Indeed, our cross-cluster hub gene analysis implicated *ARABIDOPSIS THALIANA DETOXIFICATION 1* (*AtDTX1*), which is localized in the plasma membrane of plant cells and mediates the efflux of plant-derived or exogenous toxic compounds from the cytoplasm[45]. An additional hub gene common across RcS cell populations was *CHROMATIN ASSEMBLY FACTOR 1* (*CAF-1*) *AtCAF1a*, which facilitates the incorporation of histones H3 and H4 onto newly synthesized DNA. The absence of the CAF-1 chaperone complex results in mitotic chromosome abnormalities and changes in the expression profiles of genes involved in DNA repair[46]. In addition, AtCAF1 proteins regulate mRNA deadenylation and defenses against pathogen infections[47].

To investigate the function of the RcS cell state, we compared RcS subclusters with non-RcS subclusters from the same major cell types and performed GO enrichment analysis (Supplementary Fig. 31). This, along with literature on RcS-enriched genes, revealed strong enrichment for environmental and defense response pathways. RcS subclusters showed heightened transcriptional "responsiveness," similar to the "environmentally responsive cell state" described by Oliva et al.[48] in Arabidopsis roots. These RcS cells represent highly active, plastic states.

## Recovery-induced immune responses

We found evidence for the activation of multiple immunity-related genes in the top shared hub genes among RcS subclusters. Among these genes, for example, *TETRASPANIN 8* (*TET8*) has been shown to be upregulated in response to the Flg22 and Elongation Factor Tu (EF-Tu) elicitors, which mimic pathogen infection[49]. Another shared hub gene *SALT-INDUCIBLE ZINC FINGER 1* (*SZF1*) also regulates plant immunity, as *szf1,2* knock-out mutants show increased susceptibility to *Pseudomonas syringae* pv. *tomato* DC3000 (*Pst* DC3000)[50]. Three other shared hub genes (*AT5G41750, AT5G41740,* and *AT4G19520*) encode TOLL INTERLEUKIN RECEPTOR (TIR)–type NB-LRR proteins and thus belong to the most common class of disease resistance genes in plants[51]. Another example is *ACTIVATED DISEASE RESISTANCE 2* (*ADR2*) that enhances resistance to biotrophic pathogens[52]. Finally, we also found *EARLY RESPONSIVE TO DEHYDRATION 15* (*ERD15*) as a top shared hub gene within the RcS enriched modules, which belongs to a small and highly conserved protein family that is ubiquitous but specific to the plant kingdom. Overexpression of ERD15 proteins in response to various pathogen elicitors was shown to improve resistance to known pathogens and has also been shown to impair drought tolerance[53]. Thus, the activation of a gene such as *ERD15* specifically upon rehydration may be beneficial to contribute to plant defense without compromising the drought response.

We hypothesized that the transcriptomic upregulation of immune system processes during drought recovery is an innate response to post-drought rehydration rather than a response to microbial stimulus following rehydration. To test this hypothesis, we examined the upregulation of recovery-induced genes under axenic conditions. Arabidopsis seedlings were grown on sterile agar plates for 14 days and then transferred to low-water-content agar plates[54], which we produced by decreasing the water content of the media to 50% (thus imposing moderate osmotic stress) (Supplementary Fig. 32a, b). After 14 days on the low-water-content plates, plants were rehydrated and collected at different recovery time points (0, 15, 30, 90, and 120 min) for bulk RNA-seq analysis. The upregulation of the known drought marker genes *RESPONSIVE TO DESICCATION 29 A (RD29A), RD20,* and *DELTA1-PYRROLINE-5-CARBOXYLATE SYNTHASE 1 (P5CS)*[55–57] confirmed that the plants were experiencing moderate drought stress (Supplementary Fig. 32c). We additionally found that immune-related genes were upregulated after 15 min of rehydration even in the axenic conditions (Supplementary Fig. 32d). These results support the hypothesis that plants activate a prophylactic defense response upon recovery from moderate drought, independent of pathogens in their environment.

We used data from Bjornson et al.[58] to evaluate how the genes implicated in our analysis correlated with known transcriptional responses to biotic elicitors. Bjornson et al.[58] performed transcriptomic analysis across a fine-scale time series to study rapid signaling transcriptional outputs induced by well-characterized elicitors of pattern-triggered-immunity (PTI) in Arabidopsis. Their results

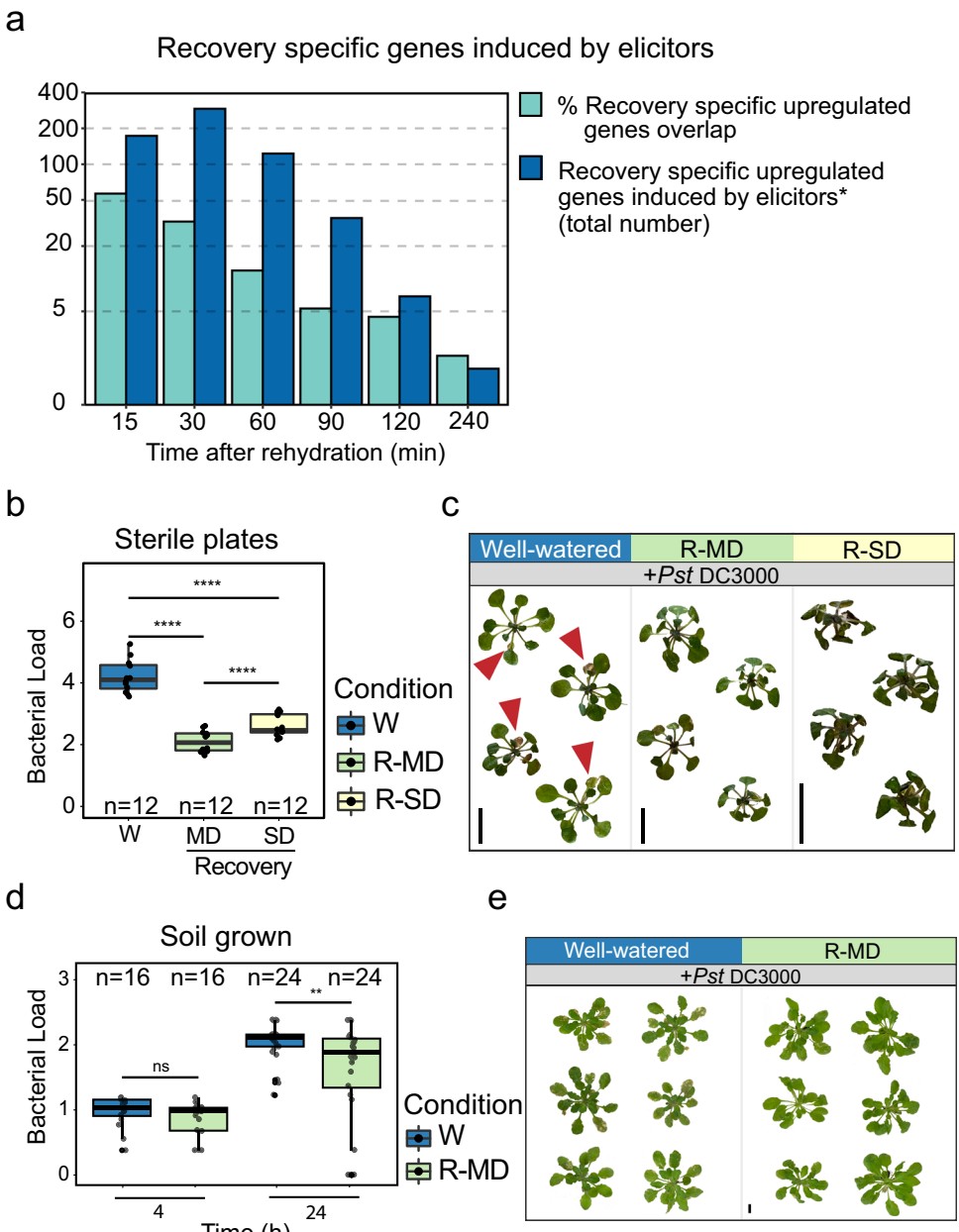

**Fig. 5 | Recovery from moderate drought enhanced pathogen resistance in *Arabidopsis*. a** Overlap between upregulated recovery-specific genes in our bulk RNA-seq data and genes upregulated by different biotic elicitors as reported by Bjornson et al. (2021). For each recovery time point, bars show both the percentage of upregulated recovery-specific genes that are also known to be activated by biotic elicitors as well as the total number. **b** Bacterial load ($log_{10}$CFU) of plants that were grown on sterile low-water-content plates and then submerged in a suspension of *Pseudomonas syringae* pv. *tomato* DC3000 (*Pst* DC3000, $OD_{600} = 0.005$) or a control solution. Bacterial growth was measured two days post-inoculation, and well-watered controls (W) were compared against plants recovering from either moderate (R-MD) or severe drought (R-SD). For this analysis, we merged two independent experiments for a total of n = 12 per treatment. Significance values were calculated with a two-way ANOVA of treatment and batch, followed by Tukey's

*post hoc* test. P-values were FDR-corrected. **c** Representative images of plants taken four days post-inoculation. Red arrowheads point out the infection symptoms. **d** Bacterial load ($log_{10}$CFU) of soil-grown plants sprayed with *Pst* DC3000 ($OD_{600} = 0.05$) or a control solution after experiencing moderate drought. Bacterial growth was measured 4 and 24 h post-inoculation and compared against controls using a two-way student's t-test. We collected 16 leaves ($n = 16$) to measure bacterial load at the initial time point, and 24 leaves (n = 24) to measure bacterial growth 24 h post infection. **e**, Representative images of plants taken 14 days post-inoculation. In **b** and **d**: ns = $P > 0.05$, * $P ≤ 0.05$, **$P ≤ 0.01$, ***$P ≤ 0.001$, ****$P ≤ 0.0001$. Boxplots middle line shows the median, the lower and upper hinges are the 25th and 75th percentile, respectively. The whiskers extend from the hinges to the most distant value within 1.5 * IQR of the hinge, where IQR is the inter-quartile range, or distance between the first and third quartiles.

showed that the transcriptional responses to diverse microbial- or plant-derived molecular patterns are highly conserved. When we aligned these identified transcriptional responses with the recovery specific genes identified by the bulk RNA-seq data from the first two hours of our drought recovery time series (Fig. 1), we found that 50% of

recovery-specific genes upregulated after 15 min of rehydration overlap with the core responses to biotic elicitors in the Bjornson et al.[58] dataset. More generally, this analysis shows a rapid and robust peak of immune-relevant gene expression that gradually decreased with time since rehydration (Fig. 5 a).

## Drought recovery-induced immunity in vivo

To test the functionality of post-recovery immune activation, we examined whether the activation of immunity-related genes during short-term recovery from moderate drought promotes pathogen resistance in vivo. We used the sterile low-water-content plate system and examined recovery from both moderate (40% water-content) and severe stress (25% water-content), as described above. Plants from the control and moderate or severe osmotic stress conditions were rehydrated for 90 min and then inoculated with *Pst* DC3000 via submersion in a bacterial suspension. Whole rosettes were collected and weighed 48 h following infection and used to quantify bacterial growth. Plants recovering from drought had significantly lower bacterial concentrations than control plants, and recovery from moderate stress suppressed bacterial growth better than recovery from severe stress (Fig. 5b, c). These results suggest that recovery from moderate drought enhances resistance to *Pst* DC3000.

To further validate these results, we grew plants on soil (non-sterile conditions) for 30 days under short-day conditions. Drought-treated plants were transferred to dry trays and dehydrated for one week down to 30% relative soil water content, while control plants continued to receive regular irrigation. We rehydrated drought-treated plants for 90 min and then sprayed the leaves of both control and recovered plants with *Pst* DC3000. Leaf discs were collected from each treatment 4 and 24 h after inoculation, surface sterilized, and then used to measure bacterial load. The bacterial load 4 h after inoculation did not differ between well-watered and drought-recovered plants. However, 24 h after inoculation, the bacterial load of drought-recovered plants was significantly lower than that of controls, indicating increased resistance to *Pst* DC3000 in rehydrated plants (Fig. 5d, e). We call this response, which was consistent across our two separate empirical tests, drought recovery-induced immunity (DRII).

In an evolutionarily conserved mechanism, ABA mediates abiotic stress tolerance and suppresses signaling of the biotic stress-related phytohormone salicylic acid (SA). Consequently, plant immunity is decreased during abiotic stresses such as drought and salt[59]. To test if ABA has a role in mediating DRII in Arabidopsis, we repeated our functional assay in *aba1-1*[60], an ABA-deficient mutant. To test if similar amounts of bacteria entered the leaf tissue upon inoculation, we collected leaf samples four hours post-inoculation, before allowing time for bacterial proliferation. In both Col-0 and *aba1-1*, there were no significant changes in the bacterial load within the leaves after initial inoculation. However, when we counted bacterial colonies five days post inoculation, there were significantly less colonies in leaves that were inoculated after 90 mins recovery following moderate drought stress in both Col-0 as seen before, and in the *aba1-1* mutant (Supplementary Fig. 33). These results suggest that the activation of DRII in Arabidopsis is independent of ABA changes upon drought recovery.

## Drought recovery increases pathogen resistance in wild (*S. pennellii*) and domesticated (*S. lycopersicum*) tomato

Given the consistency of the DRII response in Arabidopsis, we further tested whether DRII is conserved in the vegetable crop, tomato, and in response to different pathogens. First, we examined pathogen proliferation in wild tomato plants recovering from moderate drought and infected with either *Pst* DC3000 or an additional tomato pathogen *Xanthomonas perforans* strain 97-2, which causes the tomato spot disease[61]. For this test, we used the drought-tolerant tomato species *Solanum pennellii*[62]. We grew *S. pennellii* plants at 25 °C with a photoperiod of 12 h light and 12 h dark. When the plants had two to three true leaves (~4 weeks old), we exposed them to moderate drought by stopping irrigation until the soil reached 30% water content relative to saturated pots, as we had done for Arabidopsis. Drought was then alleviated by irrigating pots to saturation, and plants were infected

90 min after rehydration. Both the drought-recovered plants and well-watered controls were syringe-infiltrated with a suspension of either *X. perforans* 97-2 or *Pst* DC3000.

We first assessed disease severity in the well-watered and drought-recovered infected leaves by imaging inoculated leaves 5 days post-infection and calculating the percentage of symptomatic area relative to the entire leaf surface. Leaves inoculated 90 min after recovery from moderate drought were significantly less symptomatic than leaves from well-watered controls. To directly measure bacterial concentrations in leaves, we compared leaf samples at an initial time point 4 h after inoculation to samples collected 5 days post-inoculation. Although a similar number of bacteria entered the leaves of all plants, plants that were inoculated after recovering from drought had a significantly lower bacterial load after five days (Supplementary Fig. 34a–c). Our results were consistent across both tested pathogens: regardless of the pathogen, tomato leaves that recovered from drought exhibited lower disease severity and reduced bacterial concentrations than control plants. Our observation that short-term recovery from moderate drought increases pathogen resistance across all treatments further suggests that DRII is a preventive immune response that enhances pathogen resistance during the initial period of drought recovery.

To further investigate DRII in an agriculturally relevant plant species, we repeated our post-drought immune challenge experiment using a cultivar of domesticated tomato, *Solanum lycopersicum* cv. M82 (M82), and the same two pathogens as before (*Pst* DC3000 and *X. perforans* 97-2). Because breeders focus on traits such as fruit size, shelf life, and brix, among others, we hypothesized that the DRII response could be lost during domestication. However, M82 plants showed similar results to *S. pennellii* upon infection with either pathogen (Fig. 6 and Supplementary Fig. 34d–f), indicating that the DRII response we observed in Arabidopsis and *S. pennellii* following short-term recovery from moderate drought is retained in domesticated tomato.

## Discussion

As drought becomes an increasing threat to global food security[63,64], understanding not just drought resistance but also drought recovery is essential. While plant responses to drought have been extensively studied, the molecular basis of recovery remains underexplored. Using bulk and single-cell RNA-seq, we identified thousands of recovery-specific genes, including a subset with potential for improving crop resilience without compromising drought tolerance (e.g., Cluster 3 genes in Fig. 1e).

Our single-cell atlas revealed transcriptional plasticity across major leaf cell types and uncovered a recovery cell state that arises rapidly upon rehydration. RcS subclusters were enriched in stress- and immune-related genes, suggesting a common adaptive state. Recently, Oliva et al.[48] characterized a highly active transcriptional state that affects a subset of cells in different cell types across the developmental axis of the Arabidopsis root tip. The conserved cell state was named environmentally responsive state (ERS), since many ERS-enriched genes were previously identified as responsive to various environmental cues.

We propose that *CAMTA1*, a known calcium-regulated transcription factor, plays a role in initiating RcS. *CAMTA1* is induced during early stages of soil dehydration and our results imply that it suppresses RcS genes during the drought period. We suggest that following rehydration, CAMTA1 is removed from these target promoter,s allowing the activation of these genes by other TFs. Previous studies have shown that calcium flux can be affected by changes in water availability (both water deficit and water uptake)[65]. While transcriptomic evidence is rather strong, direct protein-level validation and functional genetic studies are needed to confirm the proposed multi-step causal pathway from RcS to DRII.

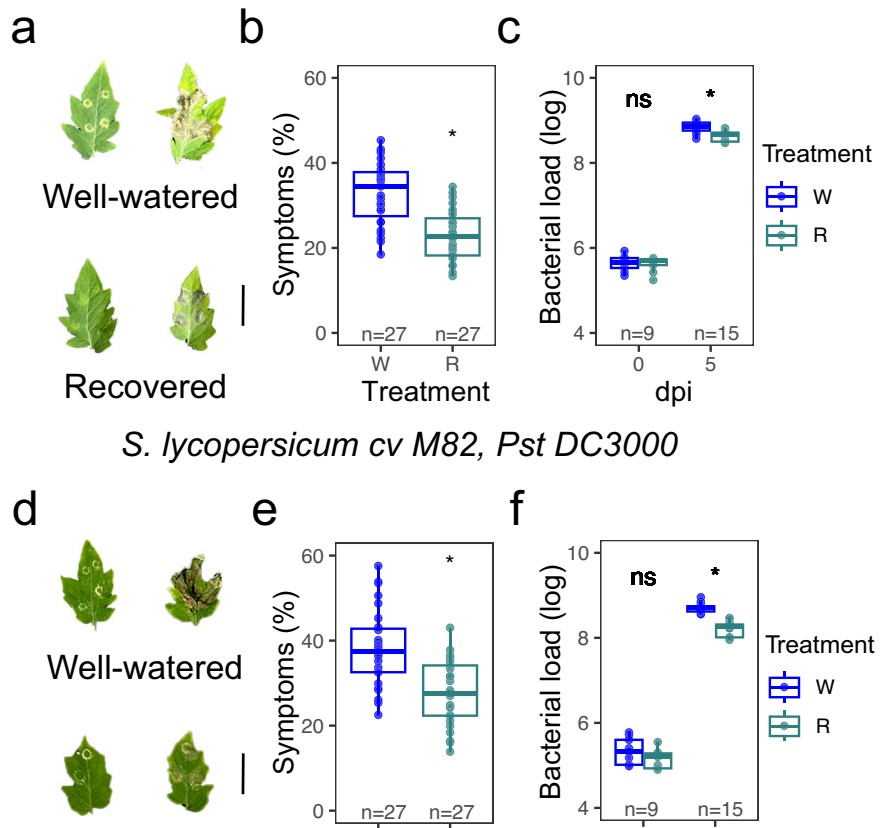

**Fig. 6 | Drought recovery-induced immunity enhanced resistance to *X. perforans* and *Pst* DC3000 in domesticated (*S. lycopersicum* cv. M82) tomato species.** Representative leaves exhibiting disease symptoms, mock (left) vs. infection (right) (**a**), symptom quantification (**b**), and bacterial load (**c**) in *Solanum lycopersicum* cv. M82 plants infected with *Pseudomonas syringae* pv. *tomato* DC3000 (*Pst DC3000*). Representative leaves exhibiting disease symptoms, mock (left) vs. infection (right) (**d**), symptom quantification (**e**), and bacterial load (**f**) in M82 plants infected with *Xanthomonas perforans*. Plants were either well-watered (W) or recovered (R) following a prior stress. Boxplots show symptom percentage (**b**, **e**) and bacterial load (log-transformed CFU per leaf area) at 0 and 5 days post-inoculation (dpi) (**c**, **f**). Each dot represents an individual biological replicate. Statistical significance was assessed using a two-sided Student's *t*-test. Significance thresholds: ns = not significant ($P > 0.05$), * $P \leq 0.05$, ** $P \leq 0.01$, *** $P \leq 0.001$, **** $P \leq 0.0001$. Boxplots indicate the median (center line), 25th and 75th percentiles (box hinges), and whiskers extending to the most extreme data points within 1.5× the interquartile range (IQR).

Importantly, our data show that drought recovery triggers a preventive immune response (DRII) that enhances resistance to bacterial pathogens. This response is ABA-independent and appears evolutionarily conserved across Arabidopsis and tomato species. Plants have evolved complex defense networks in response to microbial pathogens[51]. Allocating resources and balancing the trade-offs between biotic and abiotic stress is vital for plant survival and prosperity. Aerial pathogens like *Pst* DC3000 or *X. perforans* mainly enter plants through stomatal pores, but the stomata close in response to drought stress[66], making it harder for shoot pathogens to successfully attack plants experiencing drought. It should be noted that some pathogenic bacteria have evolved specific virulence factors to cause stomatal opening[67]. Under water-limited conditions, stomata are closed, and the environment is typically less humid, making conditions difficult for pathogens to attack plants successfully. Thus, plants dampen their immune system during drought. Moreover, plant immune responses can be sporadically induced by the circadian clock even in the absence of a pathogenic threat, a process that is correlated to the daily oscillations in humidity[68,69]. Because rehydration promotes both pathogen proliferation and stomatal opening at a time when the immune system has already been suppressed by drought/ABA, plants

are particularly vulnerable to pathogen attack during the initial stages of drought recovery[70,71]. We propose that DRII is a preventive immune response that evolved to confer resistance against pathogens during drought recovery.

In line with our findings, it was recently shown that elevated immunity can prevent pathogens from creating a niche in the apoplast via ABA manipulation[72]. The main virulence strategy to cause disease in plants by most bacterial and fungal pathogens is the induction of a water-abundant niche in the apoplast, known as water-soaking lesions[73]. During drought, ABA levels are elevated, which would normally favor the establishment of water-soaking lesions. Since plant cells contain less water due to drought conditions, the establishment of pathogens is sub-optimal. However, rapid rehydration, such as rainfall, would provide all the necessary components for pathogenesis, with ABA maintained at high levels in the leaves at the early stages of recovery.

Although plant responses to drought have been extensively studied for over a century, these findings have yet to be successfully translated to the development of drought-tolerant crops[74]. Our results highlight the presence and importance of recovery-specific mechanisms that could be targeted in future experiments and engineering

projects. We suggest a perspective for stress research in which efforts to improve crop resilience should focus not only on survival during drought but also on rapid and effective recovery. By targeting recovery-specific genes, we hope to develop crops that not only withstand drought but also recover more swiftly, ensuring minimal yield loss and sustainable food production in an era of increasing climate unpredictability.

## Methods

### Plant materials and growth conditions

*Arabidopsis thaliana* Columbia-0 (Col-0) background was used throughout this study. For the *CAMTA1* (*AT5G09410*) mutant we used SALK_008187.38.30.x, and the *aba1-1* mutant is an EMS mutant first described in Koornneef et al.[60]. Seeds were sterilized with chlorine fumes generated by mixing 100 ml bleach and 4 ml hydrochloric acid (HCl 1 M). Sterilized seeds were sown on large square petri dishes with ¾ Linsmaier & Skoog with buffer (LS) media per liter and 2% agar, and then stratified at 4 °C for three days. Plates were placed in a growth chamber and grown in short-day conditions (8 h light: 16 h dark) at 22 °C with a light intensity of ~110–130 $\mu$mol m$^{-2}$ s$^{-1}$. For drought treatments, seedlings were transferred to vermiculite pots two weeks after germination on plates, with 6 seedlings per pot. Tray size was 27.79 cm in width, 54.46 cm in length, and 6.2 cm in depth. Vermiculite was saturated with ¾ LS liquid media before the seedling transfer, 2 L media per tray. After two days, each tray was watered with 2 L ultrapure water, and this continued for two weeks until the drought treatment started. When plants were 30 days old, each pot was weighed and transferred to a dry tray. The pots in the tray were weighed daily, and the relative water content was calculated. The experiment started when the pots reached 30% relative soil-water content (SWC). Three whole Arabidopsis rosettes per treatment were collected separately as an independent sample; time 0 was 9 am (one hour after the lights turned on in the chambers). Three samples per treatment were collected alongside a well-watered control at each time point. For sterile low-water agar experiments, plates were prepared modified from Gonzales et al., 2023; 100% − 4 L DDW, 3 LS bags, 80 g agar − 120 ml per plate, 50% (moderate stress) − 2 L DDW, 3 LS bags, 80 g agar − 60 ml per plate, and 25% (severe stress) − 1 L DDW, 3 LS bags, 80 g agar − 30 ml per plate.

For the drought time course experiments, plants were grown on plates and transferred to vermiculite as described above, except that transfer from plates occurred 17 days after germination and remained on saturated vermiculite for 12 days before drought initiation. Drought was initiated by draining excess media from the pot and equilibrating to 100% relative SWC. Whole rosettes (n = 3 – 6) were sampled and frozen 4.5 h after subjective dawn each day for 5 days.

Col-0 seeds were sterilized and seeded on plates. Plates were kept in the dark at 4 C for three days. After three days, plates were moved to a growth chamber with short-day photoperiod light and 22 °C temperature.

Tomato seeds of *Solanum pennellii* and *Solanum lycopersicum* cv. M82 were germinated in a petri dish with 1/2 Murashige & Skoog (MS) with vitamins and FeNaEDTA (Cat# 07190008) media (no sucrose added). Ten-days-old seedlings were transferred to tray pots containing a mixture of vermiculite (Agrekal Moshav Habonim) and commercial soil (Tuff A.C.S) (1:1 by volume) and grown in a 12-hour light-dark cycle (12 h light: 12 h dark) at 25 °C.

### Leaf relative water content measurements

Leaf relative water content (RWC) was measured as follows: fresh leaf weight (FW) was measured immediately after leaf detachment. Leaves were then soaked for 8 h in 5 mM CaCl2 in the dark at room temperature, and the turgid weight (TW) was recorded. Dry weight (DW) was recorded after drying the leaves at 70 °C for 48 h. RWC was calculated as (FW−DW)/(TW − DW)×100.

### Stomatal conductance to water vapor measurements

Growing and drought conditions were as described above. Measurements began one hour after dawn, one leaf from 4 plants per treatment, well-watered and drought, were measured pre-rehydration (time 0 min) for Stomatal conductance ($g_{sw}$) using an LI-600 Porometer/Fluorometer (LI-COR Biosciences Inc.). Then all pots were irrigated and $g_{sw}$ of the same leaves was measured 15, 30, 60, 90, 120, 150 min after rehydration.

### Spatial transcriptomics

Spatial transcriptomics was performed using the MERSCOPE platform (Vizgen) to visualize gene expression across Arabidopsis leaf tissue under four water conditions: well-watered (time 0 and 15 min), drought, and 15 min post-rehydration. Leaf samples were collected (leaf number 7/20 from top-youngest), fixed, and processed according to the Vizgen protocol. A custom-designed probe panel targeting ~1000 genes was used, including cell type cluster markers and RcS hub genes identified from snRNA-seq analyses as well as canonical cell type markers. Tissue sections were imaged at single-molecule resolution to capture transcript localization. For this experiment, we used MERFISH 2.0 Chemistry, and the experiment was done on the Merscope Ultra Platform.

### Bacterial inoculation

*Pseudomonas syringae* pv. *tomato* (*Pst*) DC3000 was grown overnight in 10 ml liquid King's B medium containing rifampicin (40 $\mu$g/ml) (Cat#: R3501-250MG, MilliporeSigma, MA) and tetracyclin (10 mg/ml) (Cat#: T3258, MilliporeSigma, MA) at 28 °C for 24 h. Before adjusting the density, bacterial cells were washed two times with autoclaved water, followed by centrifugation at 4000 x g for 2 min and resuspension in water. For bacterial growth assays, well-watered and 90 min recovered Arabidopsis plants were inoculated with *Pst* DC3000 ($OD_{600}$ = 0.005-0.5, depending on the experiment and inoculation method). For spray treatment Silwett 77 (Cat#: S7777, PhytoTachLabs, KS) was added to bacteria before spraying. Bacterial titer was assessed as the $log_{10}$ transformed colony forming units (CFU) per plant weight when collecting whole plants from sterile plates.

For testing DRII in the absence of ABA, we collected leaf discs using a 5 mm diameter punch from leaves number 5, 6, 7, 8, and 9 at 0- and 5-days post-inoculation (dpi). Ten leaf discs were placed in a 1.5 mL Eppendorf tube and homogenized in 500 $\mu$L of 10 mM $MgCl_2$. The homogenate was serially diluted, and 10 $\mu$L of each dilution was plated on LB medium containing 50 $\mu$g mL$^{-1}$ rifampicin. The plates were incubated at 28 °C for 36 h, after which bacterial colonies were counted for titer calculation. Each experiment consisted of three or five biological replicates, for 0 dpi and 5 dpi respectively, each with three technical replicates.

For experiments performed on tomato, syringe infiltration was used for bacterial inoculation. For bacterial counting, ten leaf discs (0.5 cm in diameter) were prepared from the second and third true leaves using a hole puncher (two discs per leaf) for each sample, five samples were used for each independent experiment. Each sample was separately homogenized in 10 mM $MgCl_2$. The homogenate was serially diluted in 10 mM $MgCl_2$. 10 $\mu$l of diluted homogenate were plated on antibiotics and LB agar media on a square petri dish and incubated at 28 °C for 48 h for determination of bacterial concentrations in leaves ($log^{10}$ CFU/cm$^2$).

### Assessment of disease severity in tomato leaves

Disease severity in tomato (*S. pennellii* and *S. lycopersicum* cv. M82) leaves was assessed by calculating the percentage of the symptomatic area in the leaves relative to the whole leaf area. The calculation was done based on threshold color and color space HSB in ImageJ version 1.54 d, according to Tuang et al.[75]. Nine leaves per experiment were analyzed for the lesion percentage (%).

## RNA extraction, bulk RNA library construction

Total RNA was extracted from three independent biological replicates of each time point using RNeasy Plant Mini Kits (Cat#79254, Qiagen, CA). Tape Station checked RNA quantity for quality control. Library construction was performed using Illumina Stranded mRNA Prep (Cat#20040534, Illumina, CA).

## Nuclei extraction and single-nuclei library construction

Seedlings were transferred to vermiculite pots two weeks after germination in trays, 6 seedlings per pot. Vermiculite was saturated with ¾ LS liquid media before the seedling transfer, 2 L media per tray. After two days each tray was watered with 2 L ultra-pure water, and this continued for two weeks until the drought treatment started. When plants were 30 days old, each pot was weighed and transferred to a dry tray. The experiment began when each pot reached 30% SWC. Between 12-18 whole rosettes were collected for each time point * condition sample. We used a mortar and pestle to grind the frozen tissue. Powdered tissue was then placed in a nuclei extraction buffer [NEB; 500ul 1 M TRIS pH=7.4 (Cat# 15567027, Thermo Fisher Scientific, MA), 150 μl 1 M MgCl$_2$ (Cat# AM9530G, Fisher Scientific, MA), 100 μl 1 M NaCl (Cat# AM9760G Fisher Scientific, MA), 50 ml nuclease-free water (Cat# AM9937 Thermo Fisher Scientific, MA), 25 μl 1 M spermine (Cat# 85590-5 G, MilliporeSigma, MA), 10 μl 1 M spermidine (Cat# S2626-5G, MilliporeSigma, MA), 500 μl proteinase inhibitor (PI; Cat# P9599-5ML MilliporeSigma, MA), 250 μl BSA (Cat# B2518-100MG, MilliporeSigma, MA), 250 μl SUPERase-In (Cat# AM2696, Thermo Fisher Scientific, MA)] and incubated for 10 min. After incubation, the tissue was filtered through a 40 μm filter (Cat# 43-57040-51, pluriSelect, Germany). We then centrifuged at 500 x $g$ for 5 min at 4 °C. Supernatant was aspirated out, and nuclei were resuspended with NEB + 500 μl 10% Triton (Cat# 93443-100 ML, MilliporeSigma, MA) and no PI. We incubated for 15 min, filtered through a 40 μm filter, and spun at 500 x $g$ for 5 min. We washed until the pellet was clear. We prepared the density gradient using the Density Buffer [DB; 120 mM Tris-Cl pH=8 (Cat# AM9855G, Fisher Scientific, MA), 150 mM KCl (Cat# AM9640G, Fisher Scientific, MA), 30 mM MgCl2, 35 mL H$_2$0 per 50 mL] and filter sterilized it. We mixed 5 volumes of Optiprep (Cat# D1556-250ML, MilliporeSigma, MA) and the buffer volume to create a 50% stock. We also made a dilutant stock [400 mM Sucrose, 25 mM KCl, 5 mM MgCl$_2$, 10 mM Tris-Cl pH 8, 28 mL H$_2$0 per 50 mL] that we filter sterilized. A 45% solution was made by mixing 9 ml 50% solution and 1 ml dilutant, and a 15% solution by combining 1.5 ml 50% stock and 3.5 ml dilutant. We created a 45% solution and a 15% solution. We gently poured 2 ml of the nuclei solution at the top of the density gradient and then spun the tubes at 1500 x $g$ for 5 min with no breaks. After the spin, we pipette off nuclei and place them into a 15 ml tube with ~6 mL of NEB (with no triton or PI). We note that for vermiculite drought stressed nuclei, nuclei preparations were FACS purified. After counting the nuclei, the nuclei suspension was loaded onto microfluidic chips (10X Genomics) with HT-v3.1 chemistry to capture ~20,000 nuclei/sample. Cells were barcoded with a Chromium X Controller (10X Genomics). mRNA was reverse transcribed, and Illumina libraries were constructed for sequencing with reagents from a 3' Gene Expression HT-v3.1 kit (10X Genomics) according to the manufacturer's instructions. cDNA and final library quality were assessed using Tape-Station High Sensitivity DNA Chip (Agilent, CA).

Each of the single-nuclei samples was processed twice, to get a higher number of single nuclei transcriptomes.

## Bulk RNA-sequencing and analysis

Sequencing was performed with a NovaSeq 6000 instrument (Illumina, CA). About 40 million reads were obtained for each sample. Raw reads were processed at the IGC bioinformatics core at Salk. Alignments were performed using OSA4 and mapped to the Arabidopsis genome (TAIR 10) using Tophat2 software with default settings.

Mapped reads per library were counted using HTSeq software. For the *camta1* experiment, Reads were aligned to the TAIR10 reference genome with STAR aligner v2.5.3a, and converted to gene counts with HOMER's analyzeRepeats.pl script. Differentially expressed genes were quantified in two ways. Firstly, differentially expressed genes were identified using a spline regression model in splineTimeR v1.18.0, which were then sorted into time points using k-means clustering. Differential expression was also quantified at each time point individually using DESeq2 v1.30.0[76]. For each pairwise comparison, genes with fewer than 32 total raw counts across all samples were discarded before normalization. Genes with an absolute log$_2$foldchange > 1 and an FDR-corrected $p$-value ≤ 0.01 were pulled as significant. For functional enrichment, genes were queried for time-specific functional enrichment using over-representation analysis (ORA) in WebGestaltR v0.4.4[77]. Differentially expressed genes in each pairwise comparison were queried against the biological process non-redundant ontology, and a significance threshold of FDR-corrected $p$-value ≤ 0.05 was used.

## snRNA-seq analysis

For the snRNA-seq libraries, CellRanger (v.6.0.1) was used to perform sample-demultiplexing, barcode processing, and single-nuclei gene-UMI counting[78]. Each experiment's expression matrix was obtained by aligning to the Arabidopsis transcriptome reference (TAIR 10) using CellRanger with default parameters. For initial quality-control filtering, aligned cell and transcript counts from each treatment (well-watered, drought, recovery, 2 replicates each) were processed by Seurat (Version 4.2)[79]. The data was filtered in the following two ways: (1) Pre-filtering each replicate by removing the low-quality and outlier cells containing a high abundance of chloroplast reads (>40% of total transcripts) and mitochondrial reads (>1% of total transcripts), a low abundance of detected genes (<300 detected genes) and a relatively high abundance of unique molecular identifiers (UMIs) (>10 K for D0; >15 K and >10 K for two W15, respectively; >25 K and >20 K for two W0, respectively; >10 K for R15) (Supplementary Data 3). (2) Identifying possible doublets in each replicate using the method SCDS. SCDS implements two complementary approaches to identify doublets: one is co-expression-based doublet scoring, and the other is binary classification-based doublet scoring. Additionally, they provide a hybrid score by combining these two approaches. SCDS showed relatively high detection accuracy and computational efficiency when benchmarking with other computational methods[80]. We applied the hybrid scores for doublet estimation using R package scds() to identify likely doublet and then removed them from downstream integration. Expression data of cells passing these thresholds were log-normalized with the NormalizeData() function, and the top2K variable genes were identified with the FindVariableFeatures() function. Next, data from all conditions were integrated using Seurat's reciprocal PCA (RPCA) and FindIntegrationAnchors() functions to identify integration features and correct for potential batch effects. The integrated data were then scaled with the ScaleData() function. Principal component analysis (PCA) was carried out with the RunPCA() function, and the top 30 principal components (PCs) were retained. Clusters were identified with the FindClusters() function using the shared nearest neighbor modularity optimization with a clustering resolution set to 0.8. Clusters with only one cell were removed. This resulted in 27 initial clusters with a total of 144,494 cells. We identified a median number of 1500 genes and 3146 UMIs (representing unique transcripts), per nuclei. We detected between 25,064–26,342 genes in each sample. Cell-type identity of initial clusters was determined with canonical markers and by referring to public datasets, followed by sub-clustering each cluster using the same strategy described above.

Single-cell differential gene expression analysis was conducted using Seurat FindMarker() and FindAllmarkers() functions with the default Wilcoxon Rank Sum test. Marker genes per cell cluster and differentially expressed genes (DEGs) among conditions were both

identified by setting parameter min.pct as 0.1 and picked the top expressed markers ranking by average log₂FC. Functional enrichment analysis was carried out by over-representation analysis (ORA) using the top100 markers ranked by average log₂FC. Using the GO biological process as a reference, ORA was performed with WebGestalt[77]. Pathways with FDR < 0.05 were considered significantly enriched and visualized as plots with normalized enrichment scores. For recovery-/drought-specific DEGs, enriched known motifs were discovered with HOMER findmotifs.pl (http://homer.ucsd.edu/homer/), searching within 1000 bp upstream to 1000 kb downstream of their transcript start sites (TSSs). For the sub-clusters, recovery-enriched cell clusters were identified with >=50% cells from recovery experiment, followed with weighted gene co-expression network analysis (WGCNA) with R package hdWGCNA (https://smorabit.github.io/hdWGCNA/). Co-expression networks were constructed for each initial cell cluster with the top 20 hub genes. By checking the average expression of hub genes in sub-clusters, the recovery-specific networks were picked as those with higher expression in recovery-enriched cell clusters.

### Statistical analysis

bulk RNA-seq data, statistical analysis was performed in R using a mixed linear model function (lmer) from the package lme4 unless otherwise described. Standard errors were calculated from variance and covariance values after model fitting. The Benjamini-Hochberg method was applied for correcting of multiple testing in figures showing all pairwise comparisons of the mean estimates. For bacterial colonies count, we merged two independent experiments of plants grown on plates, a total of n = 12 per treatment. Significance values for log10(CFU) on the plates were calculated with a two-way ANOVA of treatment and batch, followed by a Tukey test. P-values are FDR-corrected. For bacterial colonies count, soil-grown plants experiment, significance values for log10(CFU) were calculated with a two-way student's t-test.

### Reporting summary

Further information on research design is available in the Nature Portfolio Reporting Summary linked to this article.

## Data availability

The sequencing data generated in this study have been deposited in the NCBI database under accession code GSE220278 [https://www.ncbi.nlm.nih.gov/geo/query/acc.cgi?acc=GSE220278]. The spatial transcriptomics data can be found in the same link, series GSE303744. The leaf relative water content, stomatal conductance to water vapor, and pathogen infection symptoms and bacterial load data generated in this study are provided in the Source Data file. Source data are provided with this paper.

## Code availability

The code used to analyze both bulk and single-nuclei RNA-Seq data is available at: https://github.com/NatanellaIE/DroughtRecovery[81]. A citable version is archived on Zenodo: https://doi.org/10.5281/zenodo.16652206.

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

## Acknowledgements
We would like to thank A. Mosquna for providing the *aba1-1* seeds. This research was supported by a Postdoctoral Award No. FI-601-2020 from the United States–Israel Binational Agricultural Research and Development Fund, granted to N.I-E. N.I.-E. is a research fellow at the George E. Hewitt Foundation for Medical Research and an Awardee of the Weizmann Institute of Science – Israel National Postdoctoral Award Program for Advancing Women in Science and the Women's Postdoctoral Career Development Award. J.R.E. is an Investigator of the Howard Hughes Medical Institute. Research reported in this publication was supported by the National Institute of General Medical Sciences of the National Institutes of Health under Award Number K99GM154136 granted to N.I-E. This work was supported by the Waitt Advanced Biophotonics Core Facility of the Salk Institute (RRID:SCR_014838) with funding from NIH-NCI CCSG P30 CA014195, NIH-NIA San Diego Nathan Shock Center P30 AG068635, The Henry L. Guenther Foundation, and the Waitt Foundation. The computational resources are partly supported by allocation MCB130189 from the Advanced Cyberinfrastructure Coordination Ecosystem: Services & Support (ACCESS) program, which the National Science Foundation supports grants #2138259, #2138286, #2138307, #2137603, and #2138296 to the Anvil HPC cluster at Purdue University.

## Author contributions
Conceptualization – N.I.-E. Data curation – N.I.-E., R.G.-C., L.P.-G., J.S., B.J., T.Z.-K., A.Y., W.O., E.O., C.B. Formal analysis – N.I.-E., K.L., J.Y., J.R.N., Z.-K.T. Funding acquisition - N.I.-E., J.R.E. Investigation - N.I.-E. Methodology - N.I.-E., T.A.L., J.S., S.B., Y.Z., T.N. Project administration - N.I.-E. Resources – J.R.E. Writing – original draft - N.I.-E., J.R.E.

## Competing interests
The authors declare no competing interests.

## Additional information

[1]Plant Biology Laboratory, The Salk Institute for Biological Studies, La Jolla, CA, USA. [2]Genomic Analysis Laboratory, The Salk Institute for Biological Studies, La Jolla, CA, USA. [3]Howard Hughes Medical Institute, The Salk Institute for Biological Studies, La Jolla, CA, USA. [4]The Razavi Newman Integrative Genomics and Bioinformatics Core Facility, The Salk Institute for Biological Studies, La Jolla, CA, USA. [5]Sanford Burnham Prebys Medical Discovery Institute, La Jolla, CA, USA. [6]Department of Plant Pathology and Microbiology, Institute of Environmental Sciences, The Robert H. Smith Faculty of Agriculture, Food and Environment, The Hebrew University of Jerusalem, Rehovot, Israel. [7]Institute of Plant Sciences and Genetics in Agriculture, The Robert H. Smith Faculty of Agriculture, Food and Environment, The Hebrew University of Jerusalem, Rehovot, Israel. [8]The Waitt Advanced Biphotonics Core Facility, The Salk Institute for Biological Studies, La Jolla, CA, USA. [9]Present address: The Sainsbury Laboratory, University of East Anglia, Norwich, UK. ✉e-mail: ecker@salk.edu

