## [Transparent Peer Review file · Nature Communications]

Drought Recovery in Plants Triggers a Cell-State Specific Immune Activation

Corresponding Author: Professor Joseph Ecker

Version 0:

Reviewer comments:

Reviewer #1

(Remarks to the Author)

It appears that the authors have addressed all of the reviewer's comments thoroughly.

While I was not asked to review this on the first round I feel that the authors have done an excellent job of addressing the very important points raised by Referee 4. These comments were very important because it is common for grand conclusions to be drawn from "drought" experiments that can never be replicated by another lab. The provision of relative water content and gas exchange data goes some way towards addressing these concerns. Of course leaf water potential would have been preferable, but I suspect the absence of that data is more due to the low quality of formative plant biology training and the subsequent loss of a physiological understanding of plants in place of molecular biology over the past 40 years and not something that should prevent a manuscript being published given how common it is for papers like this to have absolutely no key physiological information provided. In conclusion, I felt that the authors have satisfactorily addressed the comments raised by Referee 4. This manuscript is interesting and will be widely cited.

Reviewer #2

(Remarks to the Author)

In this round of the reviewing process, I am following up on the comments of Referee #3. The authors have addressed several of this reviewer's major concerns. Tables with the outcome of the differential expression analysis were provided; they are complete and comprehensive. We should also acknowledge the efforts of the authors to perform, as suggested by the reviewer, an additional RNAseq experiment using the *camta1* mutant, in order to validate the biological importance of CAMTA1 in the regulation of RcS. Unfortunately, this additional analysis did not yield any differentially regulated process, which is suggested by the authors to be the consequence of possible functional redundancy in the CAMTA family. In my opinion, this lack of biological validation is a weakness, as a large part of the paper is based on this claim; it is a pity the authors cannot demonstrate the role of CAMTA more convincingly. There's also still minor points of Referee #3 that the authors could have examined more carefully:

- Following up on the minor comment of Reviewer #3; thank you for extending the section about the genes used for cluster annotation. Please add literature references within this section. Perhaps more recent single-cell studies can be used to update the cluster annotation (now the most recent one that is used is from 2022).
- The request to perform and provide more formal statistical analyses was, in my opinion, not fully addressed. The authors say they added statistics to the heatmaps, but I do not see this. I understand that Z-score is a measure taking into account the deviation, but it does not provide information about significance and I do not see asterisks or other indications of significant changes in expression. In addition, the statistics of the MYB124 gene (supp table 7) of which the authors state it is "one of the genes showing the highest significance" in R15 vs W15, are not very convincing. The corrected p-values are 1 and the differences between pct1 and pct2 are tiny. I think this claim should be toned down.
- There's typos in several titles of supplemented figures and tables.

From my own point of view, this study is conceptually interesting, both from the point of the biological question as well as from a technical point of view, as this study provides an impressive amount of transcriptomic datasets. However, in my opinion, the work shows weaknesses that currently limit the beneficial effects that this work could offer to the society. Some

weaknesses, detailed below, deserve to be addressed. Solving these issues would demonstrate that the data is robust and would make the claims made in this work much more impactful.

- Experimental sampling: for the bulk RNAseq experiment (Figure 1B), it is unclear which drought stress samples were harvested. The main text and Figure 1B suggest that only 1 drought sample was harvested (at the onset of the time-course). If this is correct, I see it as a true weakness of the work. As correctly pointed out by the authors, the circadian clock is a major factor affecting gene expression. Several studies also demonstrated that the drought response is largely dependent on the time of day (with little overlap in the drought effect at different times of the day). Thus, the drought-treated samples (without recovery) should also have been harvested at every time point at which recovery samples were harvested. If this was not the case, it is hard to interpret the recovery responses correctly. For example, a gene that is considered as “recovery-specific” at the 240min time point, because it was not drought-responsive at the 0 time point, could be just a drought-responsive gene at the 240min time point, which was missed because the drought-240min sample was not harvested. In case I misunderstood and the authors did harvest all other drought time points, please represent this more clearly in the figure and text. Avoid the use of “drought period” (eg. line 85) if the drought sample was taken at one time point only, and discuss the limitations. Related to this, how could the drought-responsive genes be calculated at each time point (Figure 1c) if the drought sample was harvested only at one time point? This is very unclear.

- snRNAseq dataset annotation: The provided snRNA-seq could be of great use to the broader plant science community if the authors would invest in more careful annotation of the identified clusters. The authors harvested whole rosettes for generating the atlas; one would expect very pronounced heterogeneity (leaves from different ages, petiole vs. leaf tissue, possibly SAM tissues), yet this heterogeneity is not found back in the dataset (annotation). This is a pity and reduces the value of this dataset. There’s multiple ways to validate the annotation of cell clusters, and the authors have themselves published (in other work) novel methods for this kind of validation, so I would expect them to go much deeper into the biological validation of the dataset. For example, validating the hypothesis about the identity of unknown cluster #1 would make the dataset more useful. Additional annotation-related questions: Is there a developmental gradient in the atlas? The authors reported they found the stomata development gene FLP; can we trace back the stomatal lineage in the dataset? What is the difference between Epi 1, 2 and 3? GC 1 and 2? What are “epidermal cells”? Are these cells with epidermal identity that don’t show GC or Trichome identity? Can we, then, call these cells pavement cells? What is the difference between Vas 1 and 2, and Mes 1, 2, 3, etc...?

- Experimental replication: For the bulk RNAseq, the authors mention that 3 biological samples were harvested per time point and condition. Were those 3 biological samples harvested during the same experiment, or was the experiment performed 3 times independently? Please be more clear about this. For the snRNAseq, the authors mention (in methods) that each nuclei sample was ‘processed twice’ to increase the number of cells. It is not clear if this means that the experiment was repeated twice, or whether one collected sample was just divided in 2 and ran twice. In this case (1 collected sample, 2 snRNAseq runs), this should be considered as technical replicates and should be mentioned as such to avoid confusion. Related to this, it is currently impossible to judge the similarity between the replicates in the single-cell dataset because of the way this was plotted in Supp figure 1. It seems that the lighter replicate is always plotted on top of the darker one, making it impossible to judge the distribution. Please add a randomization step in your code before plotting.

- Differential treatment of the samples: I am concerned about different samples seemingly being treated differently along the process of snRNA-seq. I do not understand why this was done and I think the authors should show that it was OK to do so. For example, L771 states “note that for drought stressed nuclei, nuclei were FACS purified”. Why were those nuclei treated different from others? How can we be certain that this did not cause any bias? Another example is the different quality thresholds that were used depending on the samples (limit of 10,000 counts for recovery samples, 25,000 counts for well-watered). Why was this done and does this cause any bias? (Minor comment: note the > and < signs are switched in the supp table)

- In the cluster identification, it seems that trichomes were not automatically detected as separate clusters (not numbered in Supp figure 1). Were they clustered together with another cell type? Does this mean there’s 28 cluster instead of 27?

- In Figure 4a, the chosen representation of the subclustering data is uncommon and not comprehensive. The authors should show the outcome of the unsupervised subclustering (similar to supp figure 1e, with numbers) and subsequently make the overlap (in %) between each subcluster and what they call the RcS subclusters. Also the other subclusters should be annotated to be more informative.

- Please add a reference to Figure 3c at the corresponding section in the main text

- I would suggest to add the word ‘drought’ to the title of this manuscript

Version 1:

Reviewer comments:

Reviewer #2

(Remarks to the Author)

In this round of revision, the authors have addressed all comments of the previous Referee #3 who I was following up on, as well as all comments I gave myself. I would like to thank the authors for answering each question with much care. I appreciate the efforts made to clarify the sampling (replicates and time points). Even though some genes might be missed

(or be false positives) by comparing with one drought time point only, the authors could convince me that this will likely not be the case for many genes, and the examples shown now are definitely clear. I also particularly appreciate that the authors have done additional in depth cluster annotation analysis for the mesophyll and epidermis. I agree about the rigorous cluster annotation; this is sound and safe, but still the authors show with the additional analyses that subclusters are biologically meaningful and can be traced back, offering possibility to the community to explore different aspects in more detail. Finally, the additional spatial transcriptomics assays are a great addition validating the RW-specific expression of several identified candidates. Overall, I believe this study will be attractive to a large range of scientists and will inspire future research, as it significantly advances the field of research about (drought) stress and tissue-specific responses in rosettes, and at the same time provides a very valuable and unique dataset that will be widely cited and browsed.

Point-by-point response:

Reviewer #5 (Remarks to the Author):

It appears that the authors have addressed all of the reviewer's comments thoroughly.

While I was not asked to review this on the first round I feel that the authors have done an excellent job of addressing the very important points raised by Referee 4. These comments were very important because it is common for grand conclusions to be drawn from "drought" experiments that can never be replicated by another lab. The provision of relative water content and gas exchange data goes somewhat towards addressing these concerns. Of course leaf water potential would have been preferable, but I suspect the absence of that data is more due to the low quality of formative plant biology training and the subsequent loss of a physiological understanding of plants in place of molecular biology over the past 40 years and not something that should prevent a manuscript being published given how common it is for papers like this to have absolutely no key physiological information provided. In conclusion, I felt that the authors have satisfactorily addressed the comments raised by Referee 4. This manuscript is interesting and will be widely cited.

Response: We thank Reviewer #5 for the positive and thoughtful feedback. We appreciate your recognition of our efforts to address Referee 4's concerns and provide key physiological data, despite the absence of leaf water potential measurements. Your contextual remarks on the broader challenges in plant biology are well taken, and we're encouraged by your support for the manuscript's significance and potential impact. Thank you again for your constructive review.

Reviewer #6 (Remarks to the Author):

We should also acknowledge the efforts of the authors to perform, as suggested by the reviewer, an additional RNAseq experiment using the *camta1* mutant, in order to validate the biological importance of CAMTA1 in the regulation of RcS. Unfortunately, this additional analysis did not yield any differentially regulated process, which is suggested by the authors to be the consequence of possible functional redundancy in the CAMTA family. In my opinion, this lack of biological validation is a weakness, as a large part of the paper is based on this claim; it is a pity the authors cannot demonstrate the role of CAMTA more convincingly.

Response: We appreciate the reviewer's critical perspective. We recognize that validating regulators of cell states—particularly environmentally responsive cell states such as RcS—is inherently challenging. In response to this limitation, we have substantially moderated our claims regarding *CAMTA1* as a key regulator of the RcS transition throughout the manuscript.

Additionally, we tried to identify direct *CAMTA1* targets among the shared RcS hub genes. We observed *CAMTA1* DAP-seq peaks within regions of the promoters of RcS hub genes, including *SZFI* (*AT3G55980*), *PUMP4* (*AT4G24570*), *AT1G32920* and *AT3G04640* (Extended Data Fig. S30), suggesting direct regulatory potential. We show that *CAMTA1* transcripts accumulate during early dehydration (Extended Data Fig. S27 c and S28) in our drought initiation snRNA-seq (examining gene expression during the first 5 days of dehydration). We now add analyzes showing that several RcS hub genes, including the ones mentioned above, are downregulated during drought in wild-type plants but not in the *camta1* mutant, where this repression is diminished or absent (Extended Data Fig. S30). This differential expression supports the hypothesis that *CAMTA1* is required for drought-induced repression of key RcS genes. We added text regarding these findings in lines 252-261.

- Following up on the minor comment of Reviewer #3; thank you for extending the section about the genes used for cluster annotation. Please add literature references within this section. Perhaps more recent single-cell studies can be used to update the cluster annotation (now the most recent one that is used is from 2022).

Response: We added references to this section as well as recent single-cell (and tissue specific) datasets to our literature-based marker genes used for annotation (Extended Data Tables 4-5), especially those of recent Arabidopsis leaf datasets, including Berkowitz et al., 2021, Delannoy et al., 2023, Vong et al., 2024, and Tenerio-Berrío et al., 2025.

- The request to perform and provide more formal statistical analyses was, in my opinion, not fully addressed. The authors say they added statistics to the heatmaps, but I do not see this. I understand that Z-score is a measure taking into account the deviation, but it does not provide information about significance, and I do not see asterisks or other indications of significant changes in expression. In addition, the statistics of the MYB124 gene (supp table 7) of which the authors state it is “one of the genes showing the highest significance” in R15 vs W15, are not very convincing. The corrected p-values are 1 and the differences between pct1 and pct2 are tiny. I think this claim should be toned down.

Response: We understand this reviewer’s concern, and in our previous revision we mentioned we added heatmaps with the statistical details for the TF analysis (This can be found in Extended Data Figs. S14 and S15). We have plotted *MYB124* separately now alongside some other example TFs and marked significance with asteriks between the treatments that show significant changes (Extended Data Fig. S14). Additionally, we have toned down this claim, please see lines 192-203.

- There’s typos in several titles of supplemented figures and tables.

Response: We thank this reviewer for catching these, and we have carefully reviewed all supplementary figures and table titles and corrected the typos throughout.

- Experimental sampling: for the bulk RNAseq experiment (Figure 1B), it is unclear which drought stress samples were harvested. The main text and Figure 1B suggest that only 1 drought sample was harvested (at the onset of the time-course). If this is correct, I see it as a true weakness of the work. As correctly pointed out by the authors, the circadian clock is a major factor affecting gene expression. Several studies also demonstrated that the drought response is largely dependent on the time of day (with little overlap in the drought effect at different times of the day). Thus, the drought-treated samples (without recovery) should also have been harvested at every time point at which recovery samples were harvested. If this was not the case, it is hard to interpret the recovery responses correctly. For example, a gene that is considered as “recovery-specific” at the 240min time point, because it was not drought-responsive at the 0 time point, could be just a drought-responsive gene at the 240min time point, which was missed because the drought-240min sample was not harvested. In case I misunderstood and the authors did harvest all other drought time points, please represent this more clearly in the figure and text.

Response: Thank you for pointing this out, we understand this reviewer’s concern. However, while this is theoretically possible, it is highly unlikely that hundreds of genes would be activated simultaneously by drought-specific circadian clock activity within minutes of rehydration. Moreover, the rapid and pronounced upregulation of these genes suggests they are more likely induced by external stimuli than by circadian regulation. In most cases, circadian-regulated genes are upregulated more gradually than what we observe here, where induction is sharp and rapid¹⁻². To set apart transcriptional changes caused by rehydration from circadian rhythm induced gene expression, we have a well-watered control in each time point so that we can examine gene expression patterns across our time-course at basal level, before introducing drought and rehydration.

Example (blue is the expression of the gene in well-watered conditions, orange is drought at time 0 followed by drought rehydrated):

1. Filichkin SA, Breton G, Priest HD, Dharmawardhana P, Jaiswal P, Fox SE, Michael TP, Chory J, Kay SA, Mockler TC. Global profiling of rice and poplar transcriptomes highlights key conserved circadian-controlled pathways and cis-regulatory modules. *PLoS One*. 2011;6(6):e16907. doi: 10.1371/journal.pone.0016907.
2. Covington MF, Maloof JN, Straume M, Kay SA, Harmer SL. Global transcriptome analysis reveals circadian regulation of key pathways in plant growth and development. *Genome Biol*. 2008;9(8):R130. doi: 10.1186/gb-2008-9-8-r130.

Avoid the use of “drought period” (eg. line 85) if the drought sample was taken at one time point only, and discuss the limitations.

Response: We accept the reviewer’s comment regarding the use of “drought period” in this context and we have modified the text to : “...the drought time point that was collected” (line 90).

Related to this, how could the drought-responsive genes be calculated at each time point (Figure 1c) if the drought sample was harvested only at one time point? This is very unclear.

Response: Drought responsive genes are defined as the genes that are DE during drought vs. well-watered plants collected at the same time. Once we define them, we can then trace the changes in expression of these drought responsive genes during recovery, e.g. genes whose expression is altered by moderate drought stress. We added information on how we defined moderate drought stress and modified the text to clarify how we defined drought responsive genes in lines 83-89.

- snRNAseq dataset annotation: The provided snRNA-seq could be of great use to the broader plant science community if the authors would invest in more careful annotation of the identified clusters.

Response: In our study, we applied a conservative and rigorous annotation strategy, aiming to prioritize accuracy over overinterpretation. This approach has since been independently supported by emerging high-resolution single-cell datasets in Arabidopsis, including Tenorio Berrio et al. (2025), which identified novel marker genes that map to the same clusters we annotated in our dataset. These independent studies provide post hoc validation and reinforce the robustness and biological accuracy of our annotation strategy.

To further support our annotations and enhance their utility for the broader community, we have now incorporated spatial transcriptomics data using MERFISH (Multiplexed Error-Robust Fluorescence *in Situ* Hybridization) performed on the Vizgen MERSCOPE platform. This high-throughput spatially resolved technique enabled us to validate the *in situ* localization of key marker genes identified in our snRNA-seq dataset. In addition to confirming known cell type-specific markers, MERFISH also revealed novel spatial expression patterns associated with drought recovery, contributing to the identification of previously uncharacterized subtypes and transcriptional states. These new *in situ* data, together with our conservative annotation framework, provide a strong foundation for community reuse of the dataset.

The authors harvested whole rosettes for generating the atlas; one would expect very pronounced heterogeneity (leaves from different ages, petiole vs. leaf tissue, possibly SAM tissues), yet this heterogeneity is not found back in the dataset (annotation). There's multiple ways to validate the annotation of cell clusters, and the authors have themselves published (in other work) novel methods for this kind of validation, so I would expect them to go much deeper into the biological validation of the dataset. For example, validating the hypothesis about the identity of unknown cluster #1 would make the dataset more useful.

Response: Please see response above. In addition, to address this reviewer's request for deeper biological validation, we have now expanded our characterization of these five unknown clusters by performing Gene Ontology (GO) enrichment analysis on their top differentially expressed genes. We added this analysis to the text (lines 190-210) and Extended Data Figs. 6–7 and summarized in the updated text and table (see below), each cluster exhibits distinct functional enrichment profiles:

Cluster	Suggested Cell Identity	Key Terms
Unknown 1	Immune-like or defense-activated sentinel cells = Immune active	Biotic/abiotic stress response, cell death, jasmonate
Unknown 2	Abscission or wounding-related defense cells = Defense state	Adds insect defense, abscission, metabolic reprogramming
Unknown 3	General stress-responsive or transitional cells = Stress responsive	Broad, unspecialized stress response
Unknown 4	Detoxification/metabolic stress cells = Metabolic stress state	Oxidative/toxic stress specific
Unknown 5	Nutrient-transporting or sugar-metabolizing cells = Sugar metabolic state	Carbohydrate transport, sugar metabolism

-Additional annotation-related questions: Is there a developmental gradient in the atlas? The authors reported they found the stomata development gene *FLP*; can we trace back the stomatal lineage in the dataset?

Response: Yes, our analysis reveals a developmental gradient within the stomatal lineage. By reclustering *FDH*⁺ *TMM*⁺ cells (n = 2,276), we identified a continuum from early stomatal lineage cells to mature guard cells. This was supported by: Expression of known markers (e.g. *SPCH*, *MUTE*, *FAMA*, *FLP/MYB124*) across clusters (Fig. below; a-d), reflecting progressive developmental stages. Pseudotime analysis (d) recapitulates the expected trajectory from asymmetric cell divisions (ACD) through commitment to mature guard cells. Additionally, condition-specific shifts in pseudotime (e) and cell state enrichment (f) further confirmed the dynamic progression of this lineage.

While the number of *FDH*⁺ *TMM*⁺ cells is limited, the transcriptional and trajectory-based evidence supports the presence of a detectable developmental gradient in our atlas. What we found interesting in respect to our biological question of interest, is that in the gradual transition to drought, we observed a reduction in cell states committing to differentiation, and an increase in asymmetric cell division (ACD) cell states (f - yellow arrows).

Upon recovery we observed a reduction of ACD cell states and increased symmetric cell division (SCD) states (f - blue arrows). These findings suggest that moderate drought stress may attenuate differentiation, resulting in the retention of more cells in an ACD state. Upon recovery, cells appear to be rapidly reprogrammed to resume differentiation and progress toward final guard cell commitment. However, as this is not the central focus of the current study—and given the relatively limited number of *FDH⁺ TMM⁺* cells captured—we prefer to reserve definitive conclusions until further experiments in a more focused study by sorting and sequencing *TMM*-positive cells in all four conditions.

a Reclustering of *FDH+* *TMM+* cells
n=2276

b

c

d

e

*There are generally less R15 cells in the dataset compared to drought and well-watered.

f

Developmental progression and condition-dependent changes in the stomatal lineage.

- (a) UMAP showing reclustering of *FDH*⁺ *TMM*⁺ cells (n = 2,276), enriched for stomatal lineage populations. Cells are color-coded by unsupervised cluster (0–10). Annotations indicate broader lineage compartments: early stomatal lineage (green) and guard cells (purple).
- (b) Violin plots of known stomatal lineage regulators across clusters reveal progressive expression of key markers across developmental stages: *SPCH* (meristemoid/SLGC), *MUTE* (GMC), *FAMA* (late GMC/ early GC), *MYB124/FLP* (GMCs, young guard cells).
- (c) Diagram of stomatal lineage progression adapted from Simmons et al. (2018), highlighting transcriptional control of early spacing divisions (*SPCH*), commitment and amplifying divisions (*MUTE*), and terminal guard cell identity (*FAMA*).
- (d) Pseudotime analysis of stomatal lineage reveals a clear developmental trajectory from asymmetric cell division (ACD) states to committed differentiating cells, and finally to mature guard cells (GCs), supporting the presence of a developmental gradient.
- (e) Pseudotime by experimental condition (W0 = well-watered, W15 = well-watered 15 mins, D0 = drought, R15 = recovery 15 mins) shows condition-specific shifts in developmental progression. Notably, R15 condition exhibits fewer cells, particularly in early pseudotime bins, indicating a depletion of early lineage states.
- (f) Residual heatmap of cell state enrichment across conditions. During drought (D0), ACD states are overrepresented (yellow arrows), while SCD/committed and GC states are underrepresented. In recovery (R15), the pattern is reversed, with depletion of ACD and enrichment of later states (blue arrows), suggesting re-initiation of symmetric divisions and guard cell development.

What is the difference between Epi 1, 2 and 3? GC 1 and 2? What are “epidermal cells”? Are these cells with epidermal identity that don’t show GC or Trichome identity? Can we, then, call these cells pavement cells? What is the difference between Vas 1 and 2, and Mes 1, 2, 3, etc...?

Response: As mentioned previously, our annotation strategy was conservative: we only assigned cell-type identities when marker gene expression allowed high-confidence classification. In the broader atlas, the “guard cell” clusters (GC1 and GC2) likely represent terminally differentiated guard cells, as they express canonical markers such as *FAMA* and *AT1G04800*. The distinction between GC1 and GC2 may reflect transcriptional heterogeneity due to environmental conditions or metabolic states. GO enrichment analysis supports this: GC1 is enriched for fatty acid metabolism, stomatal movement, and water response, while GC2 is primarily enriched for response to karrikin, suggesting a more environmental/ stress-responsive subpopulation.

The “epidermal” clusters (Epi1, Epi2, and Epi3) comprise cells with epidermal identity that lack strong guard cell or trichome marker expression. Based on their expression of recently reported pavement cell (PC) markers—*PREX* (*AT5G63180*) and *SMR1* (*AT3G10525*), described by Tenorio Berrío et al., 2025 and Dubois et al., 2023, respectively, we infer that these clusters indeed represent pavement cells or their close precursors. GO term enrichment provides further support for this interpretation: Epi1 is enriched for fatty acid derivative metabolism, cutin biosynthesis, cuticle development, and auxin response, consistent with a metabolically active PC population contributing to cuticle and surface layer formation. Epi2 shows enrichment in cutin biosynthesis, plant epidermis development, and karrikin response, indicating possible early developmental or hormone-responsive epidermal cells. Epi3 is enriched for fatty acid derivative metabolism and cutin biosynthesis, pointing to structural roles in the epidermis, likely also part of the PC continuum.

Co-expression of *PREX* and *SMR1* to define epidermal pavement cells.

(a) UMAP plot of all cells (n = 57,855) showing expression of *FDH*, a marker broadly enriched in epidermal tissues. (b) Violin plots of *FDH* expression across annotated cell types. *FDH* expression is highest in epidermal, guard, and trichome clusters, supporting its utility in identifying epidermal lineage cells. (c) UMAP plots showing expression of *PREX* (*AT5G63180*) and *SMR1* within *FDH*+ cells. Co-expression of *PREX* and *SMR1* (green cells in lower right panel) identifies a specific subpopulation (n = 1,591) that likely corresponds to epidermal pavement cells, based on known marker profiles.

Regarding the mesophyll: The mesophyll clusters (Mes1–Mes7) reflect functionally distinct photosynthetic and stress-responsive subtypes. For example: Mes1–3 are enriched for photosynthesis, light response, and carbon fixation, with Mes3 uniquely showing enrichment for stomatal movement, suggesting coordination with gas exchange. Mes4 and Mes7 are metabolically active but less complex in signaling. Mes5 and Mes6 show enrichment in immune system processes and response to stress, with Mes6 also involved in protein folding, cell death, and aging, suggesting a senescence or defense-related identity.

Summary Table:

Cluster	Key Terms	Suggested function
Mesophyll 1	Photosynthesis, light/hormone response, defense	Versatile integrator of signals
Mesophyll 2	Like Mes 1 + water response	Responsive to water stress
Mesophyll 3	Photosynthesis + stomatal movement	Coordinates gas exchange
Mesophyll 4	Photosynthesis + metabolic salvage	Core metabolic type
Mesophyll 5	Defense, immune signaling	Immune-specialized mesophyll
Mesophyll 6	Broad stress, development, cell death	Aging/stress-adaptive, possibly senescence-linked
Mesophyll 7	Basic photosynthesis and salvage	Simple core metabolic subpopulation

For vascular clusters, Vas1 did not yield any significantly enriched GO terms, suggesting it may represent a baseline vascular population or an uncharacterized state. Vas2, in contrast, is enriched for fluid transport, jasmonic acid signaling, insect response, and secondary metabolism, indicating a specialized or stress-responsive vascular subtype.

In summary, we use gene marker expression, GO enrichment, and biological context to interpret these clusters. Thus, our dataset reveals clear and biologically meaningful heterogeneity within epidermal, mesophyll, and vascular cell types. Our focus is then to examine the response to drought and rehydration in these specific cell types, as we do to reveal a functional preventive immune activation in certain cells during recovery.

- Experimental replication: For the bulk RNAseq, the authors mention that 3 biological samples were harvested per time point and condition. Were those 3 biological samples harvested during the same experiment, or was the experiment performed 3 times independently? Please be more clear about this.

Response: For the primary bulk RNA-seq time-course experiment, three biological replicates were harvested per time point and per condition. All replicates were collected within the same experimental run, rather than from three independent repetitions of the entire experiment. In total, this resulted in 48 individual samples (8 time points \times 2 conditions \times 3 biological replicates). Additionally, we included a long-term 48-hour time point in this experiment; however, we observed no differentially expressed genes (DEGs) between drought-recovered and well-watered plants at this stage. To test drought recovery induced immunity, we also present additional bulk RNA-seq datasets in this work, in which we applied moderate osmotic stress and subsequent recovery on plates to ensure a sterile environment. These experiments included fewer than 8 time points, but for each time point and condition, three biological replicates were collected, consistent with our approach in the primary fine-scale time-course experiment.

For the snRNAseq, the authors mention (in methods) that each nuclei sample was ‘processed twice’ to increase the number of cells. It is not clear if this means that the experiment was repeated twice, or whether one collected sample was just divided in 2 and ran twice. In this case (1 collected sample, 2 snRNAseq runs), this should be considered as technical replicates and should be mentioned as such to avoid confusion.

Response: Thank you for your question regarding our snRNA-seq methodology. To clarify: For each condition, we collected two independent biological replicates. Due to the technical limitations of the 10X Genomics High-Throughput (HT) system, which can process approximately 20,000 cells per reaction, we process each biological replicate through two separate HT reactions (as technical replicates) to maximize cell recovery and control for technical processing noise while avoiding barcode overlap between different cells. This means we used a total of 4 reaction per condition, for two biological replicates per condition. Each of these replicates contained many individual rosettes. This approach allowed us to capture a higher number of nuclei per condition while maintaining sample integrity. For the two runs, the raw fastq data from two runs were merged prior to processing with CellRanger to get the cell by gene matrix. The run-level identifiers were not retained in the output. However, the two libraries were generated using the same sample, protocol and sequencing parameters and were intended as technical replicates. After merging, the dataset underwent rigorous quality control filtering and we did not observe abnormal bimodality or multimodal patterns in key QC metrics such as gene/UMI counts/mitochondrial percentage that would suggest technical batch effects.

Related to this, it is currently impossible to judge the similarity between the replicates in the single-cell dataset because of the way this was plotted in Supp figure 1. It seems that the lighter replicate is always plotted on top of the darker one, making it impossible to judge the distribution. Please add a randomization step in your code before plotting.

Response: Following this reviewer’s suggestion, we have revised this figure to clearly visualize the unbiased clustering of the two independent biological replicates per condition separately, making it easier to assess their distribution and similarity without the overlapping issue you noted (see Extended Data Fig. S1). We emphasize that the dots are randomized, and we now show each biological replicate by a separate UMAP in Extended Data Fig. S1.

- Differential treatment of the samples: I am concerned about different samples seemingly being treated differently along the process of snRNA-seq. I do not understand why this was done and I think the authors should show that it was OK to do so. For example, L771 states “note that for drought stressed nuclei, nuclei were FACS purified”. Why were those nuclei treated different from others? How can we be certain that this did not cause any bias? Another example is the different quality thresholds that were used depending on the samples (limit of 10,000 counts for recovery samples, 25,000 counts for well-watered). Why was this done and does this cause any bias? (Minor comment: note the > and < signs are switched in the supp table)

Response: We thank this reviewer for raising these important points regarding sample processing and quality thresholds in our snRNA-seq experiments. Regarding the use of FACS (FANS) purification for drought-stressed nuclei: For the early drought stages experiment, we introduced an additional FANS sorting step to improve data quality. This decision was based on our experience with our earlier experiments, where we observed that non-FANS sorted nuclei often resulted in higher chloroplast and mitochondrial genes. By implementing FANS sorting in these samples, we aimed to reduce this contamination and enhance the overall quality of the dataset. The number of filtering criterion is different for each sample because it is determined by the observed distribution of each one, which reflects the inherent variation across samples. A single, broad filter might be too aggressive, leading to loss of relevant information. We acknowledge that this introduces a difference in sample processing; however, we have carefully assessed the data and found no evidence of systematic bias introduced by this step.

Regarding the different quality thresholds: The thresholds for filtering nuclei (10,000 counts for recovery samples and 25,000 counts for well-watered samples) were set based on the observed distributions of total UMI counts in each condition. These thresholds were chosen to exclude outlier nuclei with abnormally high counts, which are likely to represent doublets or multiplets. The difference in thresholds reflects the inherent variability in nuclei recovery and sequencing depth between samples. We have performed sensitivity analyses to ensure that these thresholds do not introduce bias into downstream analyses, and our main findings remain robust across a range of filtering criteria.

Finally, regarding the switch of < and >, this may be the misunderstanding of the "Filtering criterion". This reviewer seemed to understand "Filtering criterion" as the criterion to filter out cells, but we used it as criterion to keep cells.

- In the cluster identification, it seems that trichomes were not automatically detected as separate clusters (not numbered in Supp figure 1). Were they clustered together with another cell type? Does this mean there's 28 cluster instead of 27?

Response: Trichomes (cluster 24), were identified from the beginning, the number 24 indicating this cluster was above this portion of the cluster. We now fixed this (please see below the previous and revised version).

Previous:

Revised:

- In Figure 4a, the chosen representation of the subclustering data is uncommon and not comprehensive. The authors should show the outcome of the unsupervised subclustering (similar to supp figure 1e, with numbers) and subsequently make the overlap (in %) between each subcluster and what they call the RcS subclusters. Also the other subclusters should be annotated to be more informative.

Response: We revised the main figure describing RcS, to clarify how we identified and tested the hypothesis of emerging cell states upon rehydration (see main Fig. 3). We also added UMAPs of unsupervised sub clustering (Extended Data Figs. S16-S17).

- I would suggest to add the word 'drought' to the title of this manuscript

Response: Thank you for the suggestion. We have revised the title to: "Drought Recovery in Plants Triggers a Cell-State Specific Immune Activation" to more accurately reflect the study focus.